# The TrkC-PTPσ complex governs synapse maturation and anxiogenic avoidance via synaptic protein phosphorylation

Husam Khaled [1,2], Zahra Ghasemi[3,10], Mai Inagaki[4,10], Kyle Patel[3], Yusuke Naito[1,5], Benjamin Feller[1,6], Nayoung Yi[1,2], Farin B Bourojeni[7], Alfred Kihoon Lee[1,5], Nicolas Chofflet[1,5], Artur Kania [5,7,8], Hidetaka Kosako [9], Masanori Tachikawa [4,11✉], Steven Connor[3,11✉] & Hideto Takahashi [1,2,5,8,11✉]

## Abstract

The precise organization of pre- and postsynaptic terminals is crucial for normal synaptic function in the brain. In addition to its canonical role as a neurotrophin-3 receptor tyrosine kinase, post-synaptic TrkC promotes excitatory synapse organization through interaction with presynaptic receptor-type tyrosine phosphatase PTPσ. To isolate the synaptic organizer function of TrkC from its role as a neurotrophin-3 receptor, we generated mice carrying TrkC point mutations that selectively abolish PTPσ binding. The excitatory synapses in mutant mice had abnormal synaptic vesicle clustering and postsynaptic density elongation, more silent synapses, and fewer active synapses, which additionally exhibited enhanced basal transmission with impaired release probability. Alongside these phenotypes, we observed aberrant synaptic protein phosphorylation, but no differences in the neurotrophin signaling pathway. Consistent with reports linking these aberrantly phosphorylated proteins to neuropsychiatric disorders, mutant TrkC knock-in mice displayed impaired social responses and increased avoidance behavior. Thus, through its regulation of synaptic protein phosphorylation, the TrkC–PTPσ complex is crucial for the maturation, but not formation, of excitatory synapses in vivo.

**Keywords** Synapse Organizer; Neurotrophin Receptor; Protein Phosphorylation; Avoidance Behavior; Social Novelty
**Subject Categories** Membranes & Trafficking; Neuroscience

## Introduction

The tropomyosin receptor kinase C (TrkC) is a receptor-type tyrosine kinase that canonically functions as a high-affinity neurotrophin receptor through binding to its ligand, neurotrophin-3 (NT-3) (Barbacid, 1994; Huang and Reichardt, 2003). This binding drives trophic signaling cascades for the promotion of neuronal precursor proliferation, and neuronal differentiation and survival (Barbacid, 1994; Huang and Reichardt, 2003) as well as synapse formation, maturation, and plasticity, influencing the development and plasticity of neuronal networks in the brain (Park and Poo, 2013). In addition to NT-3 binding, our previous in vitro studies have uncovered that postsynaptic TrkC interacts with presynaptic protein tyrosine phosphatase sigma (PTPσ), and that this trans-synaptic interaction bidirectionally promotes the differentiation of pre- and postsynaptic sites of excitatory synapses (Naito et al, 2016; Takahashi et al, 2011; Takahashi and Craig, 2013). PTPσ belongs to the type IIa receptor-type protein tyrosine phosphatases (RPTPs) and regulates several intracellular signaling pathways through its phosphatase activity (Chagnon et al, 2004; Cornejo et al, 2021; Takahashi and Craig, 2013; Tonks, 2006; Um and Ko, 2013). Furthermore, as a presynaptic adhesion molecule, PTPσ mediates excitatory presynaptic differentiation induced by extracellular interaction with multiple postsynaptic adhesion molecules, including TrkC (Naito et al, 2016; Takahashi and Craig, 2013; Um and Ko, 2013). Thus, TrkC and PTPσ act as not only signaling molecules that modulate protein phosphorylation but also key post- and presynaptic organizers, respectively, for excitatory synapse development (Cornejo et al, 2021; Naito et al, 2016; Takahashi et al, 2011; Takahashi and Craig, 2013; Um and Ko, 2013).

TrkC and PTPσ exhibit kinase and phosphatase functions, respectively, raising the possibility that they form a unique synaptic complex that could directly induce specific signaling mechanisms

[1]Synapse Development and Plasticity Research Unit, Institut de Recherches Cliniques de Montréal (IRCM), Montreal, QC H2W 1R7, Canada. [2]Department of Molecular Biology, Faculty of Medicine, Université de Montréal, Montreal, QC H3T 1J4, Canada. [3]Department of Biology, York University, Toronto, ON M3J 1P3, Canada. [4]Graduate School of Biomedical Sciences, Tokushima University, Tokushima 770-8505, Japan. [5]Integrated Program in Neuroscience, McGill University, Montreal, QC H3A 2B2, Canada. [6]Department of Neuroscience, Faculty of medicine, Université de Montréal, Montreal, QC H3T 1J4, Canada. [7]Neural Circuit Development Laboratory, Institut de Recherches Cliniques de Montréal (IRCM), Montreal, QC H2W 1R7, Canada. [8]Division of Experimental Medicine, McGill University, Montreal, QC H3A 0G4, Canada. [9]Division of Cell Signaling, Fujii Memorial Institute of Medical Sciences, Institute of Advanced Medical Sciences, Tokushima University, Tokushima 770-8503, Japan. [10]These authors contributed equally: Zahra Ghasemi, Mai Inagaki. [11]These authors contributed equally as senior authors: Masanori Tachikawa, Steven Connor, Hideto Takahashi. ✉E-mail: tachikaw@tokushima-u.ac.jp; saconnor@yorku.ca; Hideto.Takahashi@ircm.qc.ca

necessary for synapse function. However, to date, the synaptic roles of the endogenous TrkC–PTPσ complex, especially in the in vivo context, remain unclear, and the intracellular signaling mechanisms of synaptic organizing complexes in general, and the TrkC–PTPσ complex specifically, are not very well understood. Moreover, TrkC and PTPσ have been separately linked to different types of neuropsychiatric disorders including panic disorder, autism spectrum disorders (ASDs), obsessive-compulsive disorder (OCD), mood disorders and schizophrenia (Alonso et al, 2008; Armengol et al, 2002; Dierssen et al, 2006; Gratacos et al, 2001; Muinos-Gimeno et al, 2009; Otnaess et al, 2009; Takahashi and Craig, 2013; Um and Ko, 2013; Verma et al, 2008). However, how the TrkC–PTPσ complex is involved in brain function and cognition, and whether disruption of this complex is implicated in neuropsychiatric disorders remain to be elucidated.

To better understand the role of the TrkC–PTPσ complex in excitatory synapses in vivo and in brain function, it is essential to generate a mouse model that specifically lacks TrkC–PTPσ binding while maintaining TrkC-NT-3 binding. To do so, we took advantage of a recent structural study (Coles et al, 2014) that revealed the binding interfaces between TrkC and PTPσ and determined that two amino acid residues (aspartic acid (D)240 and D242) located in the first immunoglobulin-like domain (Ig1) are necessary for TrkC–PTPσ binding and irrelevant for TrkC-NT-3 binding (Coles et al, 2014; Urfer et al, 1998; Urfer et al, 1995). The substitution of these aspartic acid residues to alanine (D240A;D242A) completely inhibits TrkC–PTPσ binding and the synaptogenic activity of TrkC (Coles et al, 2014) without affecting TrkC-NT-3 binding (Appendix Fig. S1). We therefore generated a TrkC knock-in (KI) mutant mouse line that expresses TrkC possessing the D240A;D242A point mutations and validated that this results in loss of interaction between the mutant TrkC and PTPσ in the KI homozygous mice (TrkC KI mice). We then characterized the TrkC KI mice comprehensively using structural and functional assessments of excitatory synapses, unbiased proteomic and phosphoproteomic analyses, and behavioral experiments. We found an unexpected mechanism of the TrkC–PTPσ complex in excitatory synapse organization in vivo and implicated it in several neuropsychiatric disorders.

## Results

### Generation and validation of a TrkC knock-in mutant mouse line with point mutations that abolish endogenous TrkC–PTPσ interaction

We used traditional homologous recombination to generate a mutant mouse line bearing substitutions in TrkC (D240A;D242A) that abolish PTPσ binding (Fig. 1A). We confirmed the expected homologous recombination of the targeting vector in the first generation (F1) heterozygous mice by Southern blotting using both 5′ and 3′ probes and excluded random insertion of the targeting vector by using a probe that detects the neomycin cassette (neo probe) (Fig. 1B). F1 heterozygous mice were crossed with B6 background mice expressing FLPe recombinase in germline cells (B6;SJL-Tg(ACTFLPe)9205Dym/J) (Rodriguez et al, 2000) to remove the neomycin cassette, which is flanked by two FRT sites (Fig. 1A,C). Their progeny were then crossed with C57Bl/6J mice to obtain mice lacking the ACTFLPe transgene (selected by PCR), which were backcrossed at least six times, with

confirmation of the mutated nucleotide sequences encoding D240A;D242A by DNA sequencing (Fig. 1D) to obtain the mice used for experiments.

To confirm whether TrkC KI mice lack TrkC–PTPσ interaction at an endogenous protein level, we performed co-immunoprecipitation (co-IP) experiments (Fig. 1E). TrkC has two major alternative splicing isoforms: one with and one without the intracellular tyrosine kinase (TK) domain (TrkC TK+ and TrkC TK−, respectively) (Barbacid, 1994; Huang and Reichardt, 2003; Naito et al, 2016). TrkC TK+ and TK− isoforms have identical extracellular regions, including the leucine-rich repeat (LRR) and Ig1 domains that are responsible for PTPσ binding (Ammendrup-Johnsen et al, 2015; Naito et al, 2016; Takahashi et al, 2011). Both TrkC TK+ and TrkC TK− immunoprecipitated equally with the PTPσ antibody in wild-type (WT) mouse brain samples (Fig. 1E), supportive of an endogenous protein interaction between TrkC and PTPσ. In contrast, in the brain samples from TrkC KI mice, neither TrkC TK+ nor TrkC TK− immunoprecipitated with the PTPσ antibody (Fig. 1E). Next, we investigated synaptic expression of TrkC, PTPσ and NT-3 using hippocampal synaptosome samples and found that TrkC KI and WT mice showed no significant changes in expression of TrkC TK+, TrkC TK−, PTPσ or NT-3 (Fig. 1F,G). These data indicate that knock-in of the D240A;D242A substitution abolishes endogenous TrkC–PTPσ interaction in the brain without affecting the synaptic expression of TrkC, PTPσ or NT-3.

TrkC KI mice were viable and exhibited no obvious anatomical defects (Fig. 1H). The genotype for the TrkC KI mouse line showed normal Mendelian distribution (25.7%, 50.2%, and 24.2% WT, heterozygous, and homozygous mice, respectively, 1489 mice) and equal sex distribution. Furthermore, the gross morphology, size, and weight of TrkC KI brains were normal (Fig. 1I). Nissl staining of brain sections from TrkC KI and WT littermates revealed no gross developmental defects in TrkC KI brains (Fig. 1J). Specifically, the size and structure of the hippocampus and the laminar structure of the cortex in TrkC KI mice seemed comparable to those of WT littermates (Fig. 1J). This lack of obvious brain abnormalities suggests that the in vivo abolishment of endogenous TrkC–PTPσ interaction by knock-in of TrkC D240A;D242A does not affect the canonical trophic role of TrkC as an NT-3 receptor in neuronal and brain development.

### Abnormal synaptic vesicle clustering and elongation of the postsynaptic density in excitatory synapses caused by the disruption of TrkC–PTPσ complex

Previous in vitro studies have shown that the TrkC–PTPσ trans-synaptic complex has bidirectional synaptogenic activity to promote both pre- and postsynaptic development of excitatory synapses in cultured hippocampal neurons (Ammendrup-Johnsen et al, 2015; Han et al, 2016; Naito et al, 2016; Takahashi et al, 2011). To test whether loss of TrkC–PTPσ interaction diminishes excitatory synapse development in vivo, we first performed immunohistochemistry (IHC) for excitatory synapses (Fig. 2A–C). Given the relatively high expression of TrkC proteins in the hippocampus for both WT and KI mice (Appendix Fig. S2A), we co-immunostained hippocampal cryostat sections for the vesicular glutamate transporter VGLUT1 and the postsynaptic density protein PSD-95, markers of excitatory pre- and postsynaptic sites, respectively (Fig. 2A). Surprisingly, the intensity of VGLUT1 and

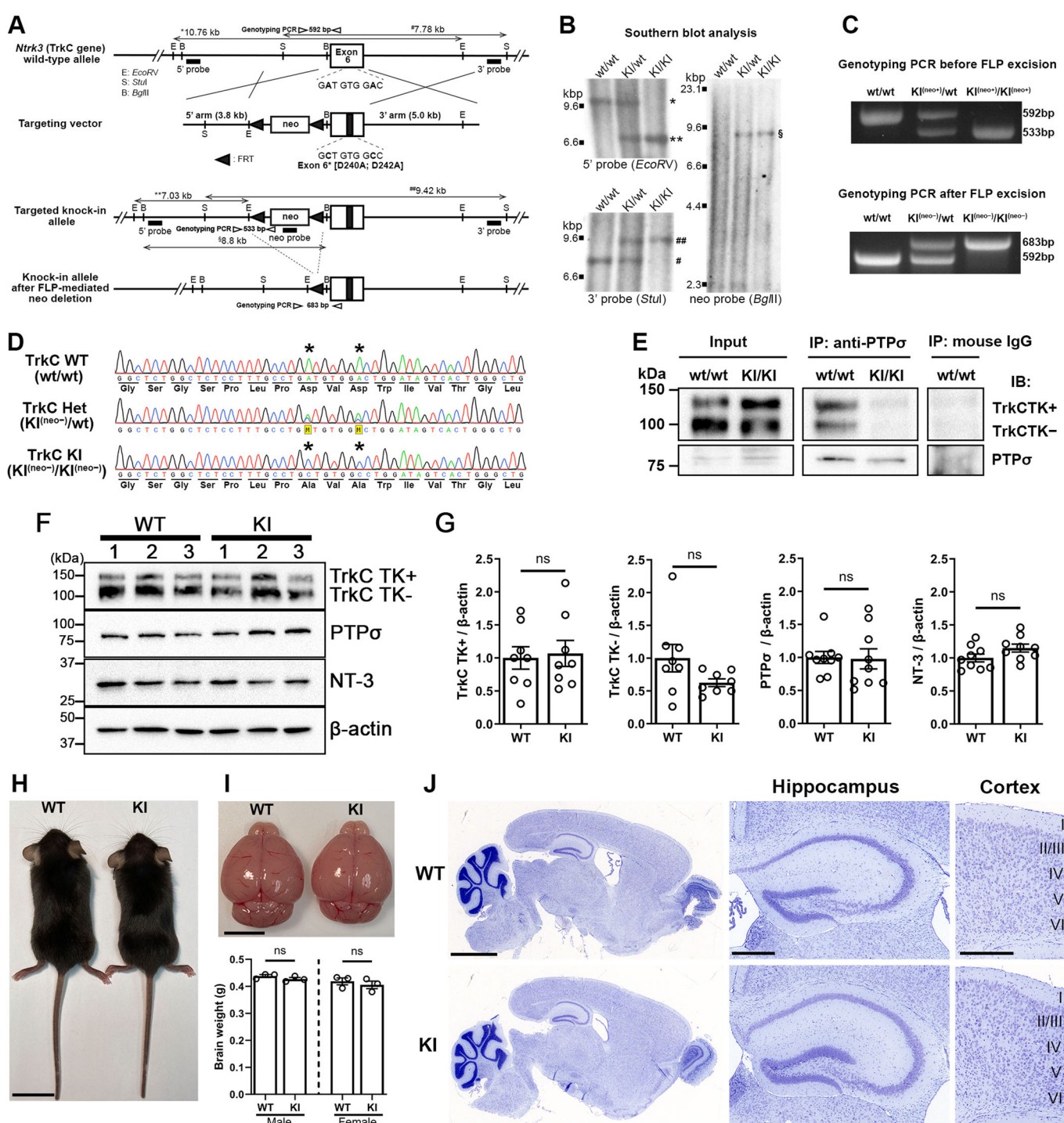

PSD-95 staining appeared stronger in TrkC KI mice (Fig. 2A). Upon quantifying the signals in the stratum radiatum (st. rad.) and stratum oriens (st. ori.) of the CA1 region, where endogenous TrkC proteins are highly expressed (Appendix Fig. S2B), we detected a significant increase in immunoreactivity for VGLUT1 and PSD-95 in the st. rad (Fig. 2B; Appendix Fig. S3) and for VGLUT1 in the st. ori of TrkC KI mice (Fig. 2C; Appendix Fig. S3). Meanwhile, we detected no change in the PSD-95 intensity in the st. ori in the KI mice (Fig. 2C; Appendix Fig. S3). There was no change in cell

number in the CA1 pyramidal cell layer, eliminating this as a possible explanation for the increased VGLUT1 and PSD-95 signals (Fig. 2D). To test whether this surprising result is due to a difference between in vitro and in vivo conditions, we performed co-immunostaining for VGLUT1 and PSD-95 in primary cultured hippocampal neurons derived from TrkC KI and WT littermates. We found a similar phenotype as in the IHC, an increase in VGLUT1 puncta intensity (Appendix Fig. S4). We further detected no change in the number of VGLUT1-positive PSD-95 puncta,

**Figure 1.   Generation and validation of the D240A;D242A TrkC knock-in mutant mouse line.**

(A) Gene targeting strategy based on a traditional homologous recombination method. (B) Validation of planned homologous recombination and lack of random insertion by Southern blotting using 5′, 3′ and neo probes. A single asterisk and hash indicate bands from the wild-type (WT) allele, whereas double ones indicate bands from the knock-in (KI) allele. (C) PCR-based genotyping before and after the removal of neo cassette by crossing with *ACTB:FLPe B6J* mice. (D) Validation of nucleotide sequences encoding D240A;D242A in TrkC heterozygous and homozygous mutant mice. The highlighted M's indicate the presence of both adenine and cytosine bases in heterozygous mice. (E) Co-immunoprecipitation validating absence of protein interaction between endogenous TrkC and PTPσ in TrkC KI homozygous mouse brains. (F) Representative immunoblots of TrkC TK + , TrkC TK − , PTPσ and NT-3 in hippocampal synaptosomes of TrkC KI mice and WT littermates. (G) Quantification of protein expression normalized to β-actin loading controls in hippocampal synaptosomes, expressed as a ratio of the values in WT mice. $n = 8$ pooled hippocampal synaptosome samples; Student's $t$ tests, $P = 0.80$, $P = 0.11$, $P = 0.90$, and $P = 0.096$ for TrkC TK + , TrkC TK − , PTPσ and NT-3, respectively. (H) No obvious anatomical defects in TrkC KI homozygous mice (TrkC KI mice). (I) No significant difference between TrkC KI and WT mouse brain size, gross morphology or weight. $n = 3$ mice per genotype per each sex. Student's $t$ tests, $P = 0.23$ for males and $P = 0.53$ for females. (J) Gross brain structure and layer formation of the hippocampus and cortex in Nissl-stained brain sections from a TrkC KI mouse and WT littermate are indistinguishable. Scale bars: 2 cm (H), 0.5 cm (I), 0.2 mm (J, left), and 500 μm (J, center and right). Data are presented as mean ± SEM. ns not significant. Source data are available online for this figure.

suggesting that the TrkC KI cultured neurons do not exhibit a reduction in excitatory synapses (Appendix Fig. S4). However, western blotting (WB) showed that expression of VGLUT1 and PSD-95 was comparable in synaptosomal and total lysate samples from the hippocampus of TrkC KI and WT littermates (Fig. 2E,F). Taken together, these data suggest that loss of the TrkC–PTPσ interaction by TrkC D240A;D242A substitution may affect the distribution of excitatory synaptic proteins without affecting their overall synaptic expression.

To further investigate structural changes of excitatory synapses in the TrkC KI mice, we used transmission electron microscopy (EM) to perform ultrastructural analysis of hippocampal CA1 synapses (Fig. 3). We first assessed asymmetric and symmetric synapse density in the st. rad. and the st. ori, with asymmetric synapses (excitatory) being defined as those containing a post-synaptic density (PSD) structure, the electron-dense lamina beneath the postsynaptic membrane, and symmetric synapses (inhibitory) being defined as those containing no PSD structure (Harris and Weinberg, 2012). We detected no significant change in the number of either asymmetric or symmetric synapses in either region in TrkC KI mice (Fig. 3A,B; Appendix Fig. S5). Given the increased PSD-95 intensity in the st. rad. but not in the st. ori. observed by IHC (Fig. 2A–C), we next quantified the length of the PSD in all imaged asymmetric synapses and found that it was significantly longer in the st. rad., but not in the st. ori., in TrkC KI mice (Fig. 3C). Thus, the PSD phenotype in the EM analysis is consistent with the st. rad-specific increase in PSD-95 signal observed in the IHC experiments (Fig. 2B,C). Next, we measured the number of synaptic vesicles (SVs) per presynaptic bouton of asymmetric synapses in the st. rad. and in the st. ori. Through analyzing asymmetric synapses of comparable size to exclude the effects of synapse size on SV number per bouton, we found that the asymmetric synapses of TrkC KI and WT mice contained a comparable number of SVs per bouton area (Fig. 3D,E). However, SVs seemed distributed more densely and more clustered in presynaptic boutons of many TrkC KI asymmetric synapses (Fig. 3D), and the distance between SVs was indeed significantly smaller in TrkC KI synapses than in WT synapses (Fig. 3F). These EM data indicate that the loss of TrkC–PTPσ interaction abnormally facilitates the clustering of SVs in presynaptic boutons without affecting SV number per bouton. Such enhanced SV clustering could also lead to the increased VGLUT1 signal intensity observed with IHC. Thus, the loss of TrkC–PTPσ interaction causes abnormal PSD elongation in the hippocampal CA1 st. rad. as well

as aberrant SV clustering in excitatory synapses in the st. rad. and the st. ori. without affecting the number of excitatory synapses in these regions. These results suggest that loss of TrkC–PTPσ interaction may affect excitatory synapse maturation rather than formation.

## Loss of TrkC–PTPσ interaction increases silent synapses and induces aberrant active synapses that show enhanced basal transmission with impaired release probability

We next carried out functional assessments of excitatory synapses in acute brain slice electrophysiological experiments. We first performed whole-cell patch-clamp recording of evoked excitatory postsynaptic currents (EPSCs) at Schaffer collateral (SC)-CA1 pyramidal cell synapses to measure the ratio of α-amino-3-hydroxy-5-methyl-4-isoxazole propionic acid glutamate receptor (AMPAR) to N-methyl-D-aspartate glutamate receptor (NMDAR) currents. The AMPAR/NMDAR ratio of TrkC KI SC-CA1 synapses was significantly lower than that of WT ones (Fig. 4A,B). Given the lower AMPAR/NMDAR ratio and the apparent synaptic disorganization phenotype observed in Fig. 3, which could be indicative of compromised synapse development, we next probed for changes in "silent synapses". Silent synapses are synapses that harbor NMDARs but not AMPARs, which are more abundant early in brain development (Isaac et al, 1995; Liao et al, 1995), and provide a functional marker for altered synapse development. To measure silent synapses, we analyzed the ratio of the coefficient of variance (CV) of NMDAR (CV-NMDAR) to CV of AMPAR (CV-AMPAR) currents (Kullmann, 1994; Manabe et al, 1993) and found that it was significantly reduced in TrkC KI synapses (Fig. 4C). Given that there is no change in total excitatory synapse number in the st. rad (Fig. 3A,B), these results suggest that there are more silent synapses and fewer active synapses (NMDAR- and AMPAR-positive synapses) in TrkC KI mice than in WT mice.

To assess the functional properties of active synapses, we next recorded AMPAR-mediated miniature EPSCs (mEPSCs) from hippocampal CA1 pyramidal cells. Curiously, mEPSC frequency was significantly increased in TrkC KI mice relative to WT mice (Fig. 4D,E). On the other hand, there was no difference in mEPSC amplitude between TrkC KI and WT mice (Fig. 4D,F). These data suggest that active synapses in TrkC KI mice may contain increased release sites and/or increased release probability without changing postsynaptic AMPARs.

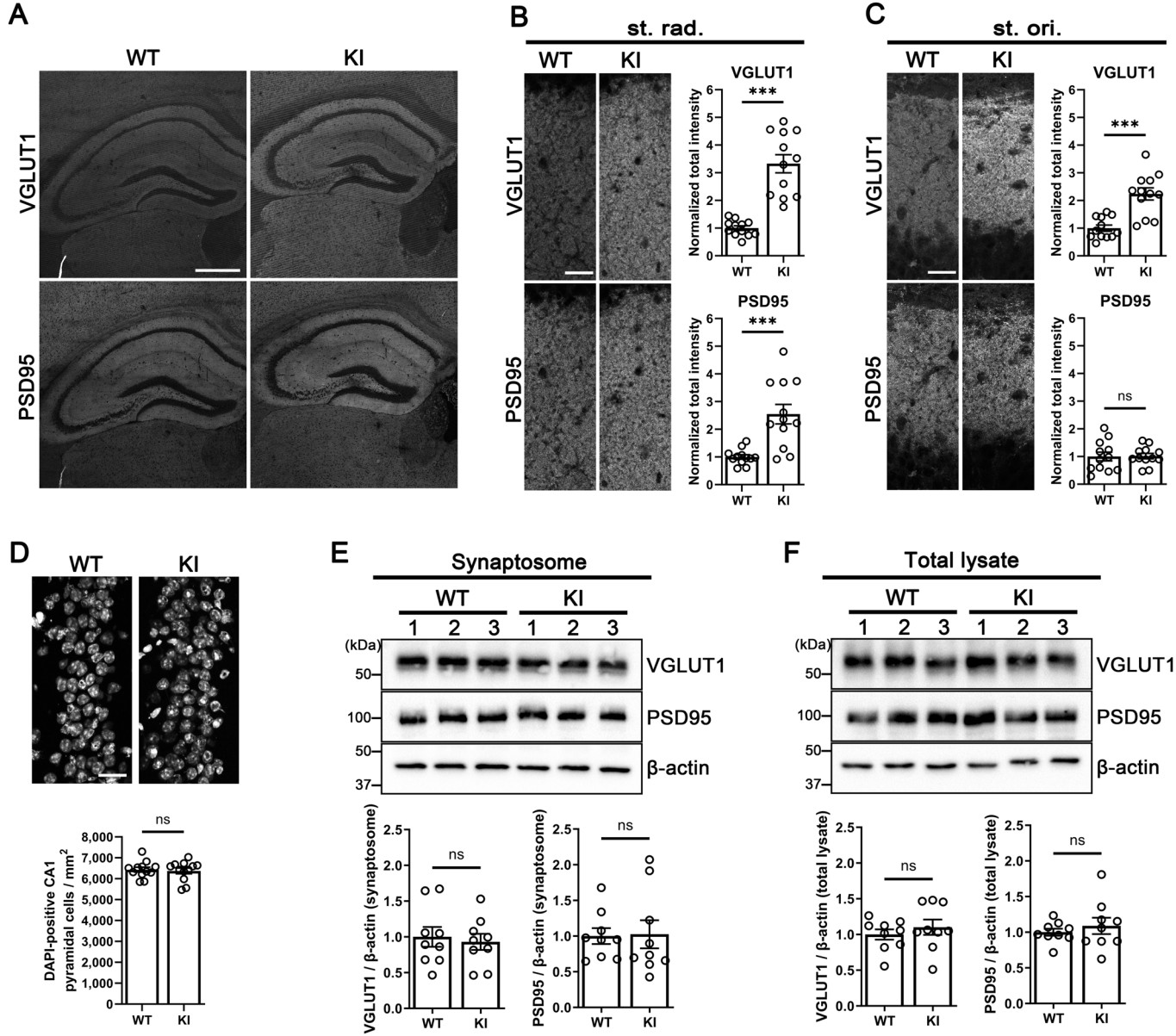

**Figure 2.  Synaptic expression of excitatory synaptic proteins VGLUT1 and PSD-95 is unchanged but their immunofluorescence signal intensity is increased in TrkC KI mice.**

(A) Double immunolabeling of a brain section for VGLUT1 and PSD-95 showing the hippocampal CA1 region of TrkC KI mice and WT littermates. (B) High-magnification images of VGLUT1 and PSD-95 immunolabeling in the stratum radiatum (st. rad.) and quantification of the total intensity of VGLUT1 and PSD-95 puncta per area in this region. $n = 12$ regions from four mice (3 regions from each mouse) per genotype. Student's $t$ tests, ***$P = 7.0 \times 10^{-7}$ for VGLUT1 and ***$P = 0.00030$ for PSD-95. (C) High-magnification images of VGLUT1 and PSD-95 immunolabeling in the stratum oriens (st. ori.) and quantification of the total intensity of VGLUT1 and PSD-95 puncta per area in this region. $n = 12$ regions from 4 mice per genotype. Student's $t$ tests, ***$P = 5.0 \times 10^{-5}$ for VGLUT1 and $P = 0.93$ for PSD-95. (D) High-magnification images and quantification of the number of DAPI-labeled cells per area in the hippocampal CA1 pyramidal layer. $n = 12$ regions from 4 mice per genotype. Student's $t$ test, $P = 0.74$. (E) Representative immunoblots of VGLUT1 and PSD-95 in synaptosomes from the hippocampus of TrkC KI mice and WT littermates, with quantification showing protein expression normalized to β-actin loading controls and expressed as a ratio of the values in WT mice. $n = 9$ hippocampal synaptosomal samples per genotype, with each sample prepared from pooled hippocampi derived from 6 mice of the same genotype. Student's $t$ tests, $P = 0.70$ for VGLUT1 and $P = 0.91$ for PSD-95. (F) Representative immunoblots of VGLUT1 and PSD-95 in total lysates from the hippocampus of TrkC KI mice and WT littermates, with quantification showing protein expression normalized to β-actin loading controls and expressed as a ratio of the values in WT mice. $n = 9$ hippocampal total lysate samples per genotype, with each sample prepared from pooled hippocampi derived from six mice of the same genotype. Student's $t$ tests, $P = 0.45$ for VGLUT1 and $P = 0.49$ for PSD-95. Scale bars: 500 μm (A) and 25 μm (B–D). Data are presented as mean ± SEM. ns not significant. Source data are available online for this figure.

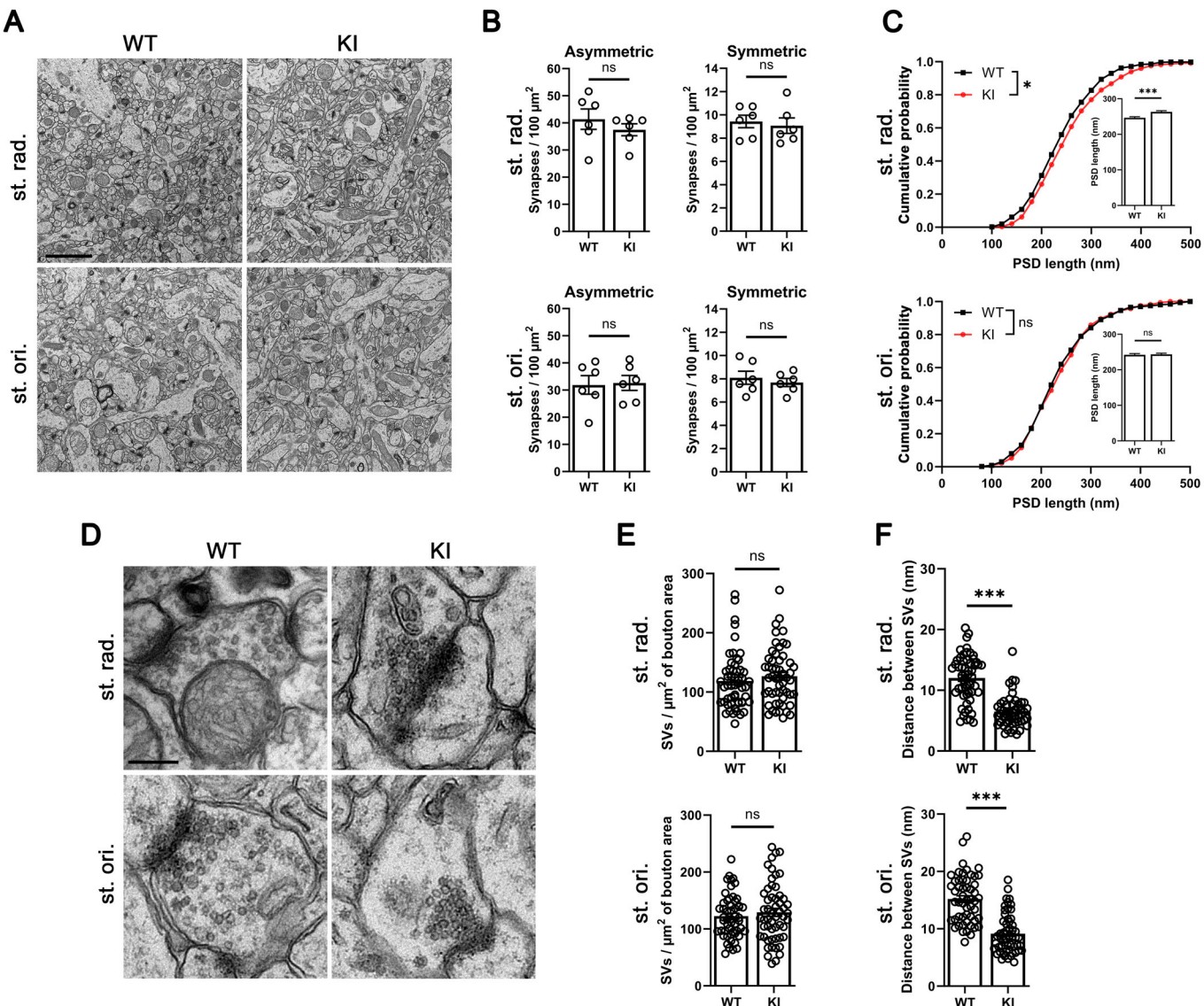

**Figure 3.  Excitatory synapses in TrkC KI mice show condensed clustering of synaptic vesicles and postsynaptic density elongation.**

(A) Electron micrographs showing the hippocampal CA1 st. rad. and st. ori. of TrkC KI mice and WT littermates at 4 postnatal weeks. (B) Quantification of the number of asymmetric and symmetric synapses in the st. rad. and the st. ori. $n = 6$ samples from three mice for each genotype. Student's $t$ test, $P = 0.40$ and $P = 0.68$ for asymmetric and symmetric synapses in the st. rad., and $P = 0.87$ and $P = 0.55$ for asymmetric and symmetric synapses in the st. ori., respectively. (C) Quantification of PSD length in asymmetric synapses in the st. rad. and the st. ori. in TrkC KI mice and WT littermates. $n = 556$ WT and 566 KI asymmetric synapses in the st. rad. and 466 WT and 493 KI synapses in the st. ori. from three mice for each genotype were analyzed. Kolmogorov–Smirnov test, *$P = 0.043$ and $P = 0.24$ in the st. rad. and st. ori cumulative probability curves, respectively. Student's $t$ tests, ***$P = 0.00014$ and $P = 0.77$ in the st. rad. and st. ori. bar graphs, respectively. (D) Electron micrographs showing asymmetric synapses on dendritic regions in the hippocampal CA1 st. rad and st. ori. of TrkC KI and WT mice. (E, F) Quantification of the number of synaptic vesicles (SVs) per bouton area (E) and distance between SVs in asymmetric synapses (F) in the st. rad. and st. ori. $n = 54$ synapses for each region from three mice for each genotype. Student's $t$ tests, $P = 0.38$ and $P = 0.44$ in the st. rad. and st. ori., respectively, for SV number (E), and ***$P = 3.8 \times 10^{-14}$ and ***$P = 4.5 \times 10^{-13}$ in the st. rad. and st. ori., respectively, for SV distance (F). Scale bars represent 2 µm (A) and 250 nm (D). Data are presented as mean ± SEM. ns not significant. Source data are available online for this figure.

Next, to assess basal evoked synaptic transmission, we performed extracellular recording of field excitatory postsynaptic potentials (fEPSPs) at SC-CA1 pyramidal cell synapses in the hippocampus. An input/output (I/O) curve was established by applying stepwise increases in electrical stimulation to axons comprising the SC pathway while recording fEPSPs within the CA1 st. rad. (Fig. 4G,H). Significant increases in fEPSP slopes against

both stimulus intensity and fiber volley amplitude were observed in TrkC KI hippocampal slices (Fig. 4G,H). These data indicate that basal evoked synaptic transmission is enhanced in TrkC KI SC-CA1 synapses. Comparisons of raw fiber volley amplitudes further suggest that these changes are not due to altered properties of SC axons (Fig. 4I). We further investigated presynaptic glutamate release probability by measuring paired-pulse facilitation (PPF), a

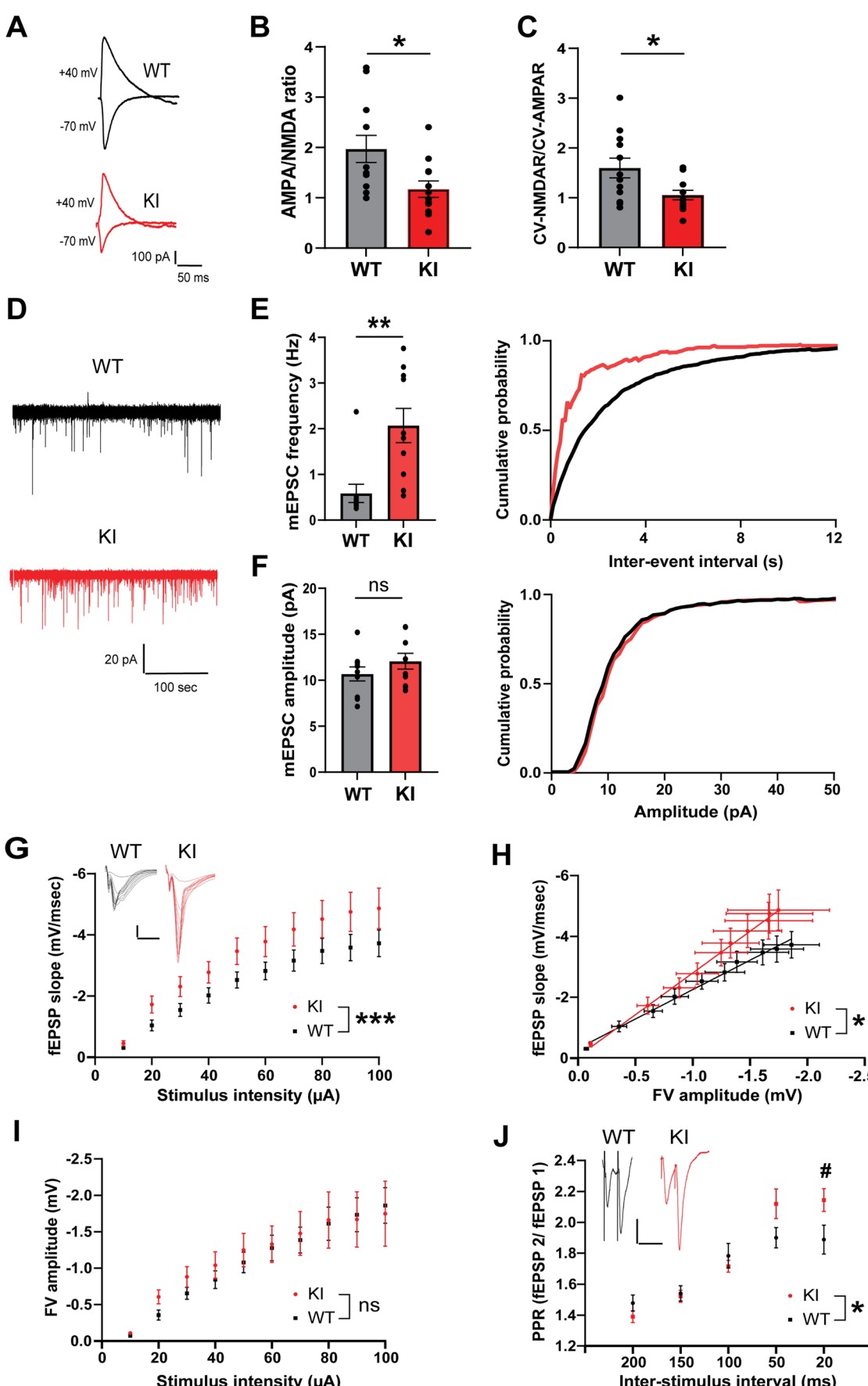

**Figure 4. Preventing TrkC–PTPσ interaction increases silent synapses and causes aberrant active synapses on hippocampal CA1 pyramidal cells.**

(A) Representative traces of AMPAR-mediated evoked EPSCs recorded at $-70$ mV and AMPAR plus NMDAR-mediated evoked EPSCs recorded at $+40$ mV from CA1 pyramidal cells of TrkC KI (red) and WT (black) mice in response to Schaffer collateral stimulation. (B) A comparison of the AMPAR/NMDAR ratio showed a significant reduction in TrkC KI mice. Student's $t$ test, $*P = 0.019$, $n = 12$ cells/5 mice per group. (C) The ratio of CV-NMDAR to CV-AMPAR significantly decreased in TrkC KI compared to WT mice. Student's $t$ test, $*P = 0.022$, $n = 12$ cells/5 mice per group. (D) Sample traces of AMPAR-mediated mEPSCs recorded from hippocampal CA1 pyramidal cells in TrkC KI (red) and WT (black) mice. (E) mEPSC frequency was increased in TrkC KI compared to WT mice. Student's $t$ test, $**P = 0.0026$, $n = 10$ WT cells/5 mice and 10 KI cells/4 mice. This increase resulted in a leftward shift in the cumulative probability curve corresponding to decreased inter-event intervals (Kolmogorov–Smirnov test, $P < 1.0 \times 10^{-15}$). (F) No significant differences were found between TrkC KI and WT neurons in mEPSC amplitude. Student's $t$ test, $P = 0.24$, $n = 10$ WT cells/5 mice and 10 KI cells/4 mice. (G) Input–output curves of fEPSPs elicited by Schaffer collateral stimulation and recorded from the CA1 st. rad. fEPSP input/output responses were significantly elevated in TrkC KI mouse hippocampal slices. $n = 7$ WT slices (3 mice) and 8 TrkC KI slices (3 mice); two-way repeated measures ANOVA, $F_{(1,130)} = 22.34$, $***P = 5.8 \times 10^{-6}$. Inset: Representative fEPSPs from CA1 elicited with stepwise increases in stimulation intensity (scale bar, 1 mV, 5 ms). (H) Linear fit slopes comparing fiber volley (FV) to fEPSP amplitude plots. The slopes were significantly different ($F_{(1,146)} = 6.221$, $*P = 0.014$), with FV/slope values elevated in the TrkC KI slices. (I) Quantification of FV amplitude. No differences were detected in the relationships between stimulus intensity and presynaptic FV amplitude between groups. Two-way repeated measures ANOVA, $F_{(1,130)} = 0.7274$, $P = 0.40$. (J) A presynaptic function assay identified a significant increase in the paired-pulse ratio (PPR) at inter-stimulus intervals <50 ms. $n = 8$ WT slices (5 mice) and 8 KI slices (6 mice); two-way repeated measures ANOVA, $F_{(4,70)} = 3.121$, $*P = 0.020$; Šídák's post hoc test at 20 ms ($^{\#}P = 0.039$). Inset: fEPSP PPR trace at 20 ms interpulse interval; scale bar, 1 mV, 40 ms. Data are presented as mean ± SEM. ns not significant. Source data are available online for this figure.

form of short-term presynaptic plasticity (Jackman and Regehr, 2017). Notably, significant increases in PPF at interpulse intervals below 50 msec were detected in TrkC KI mice (Fig. 4J), suggesting that TrkC KI SC-CA1 synapses exhibit decreased evoked presynaptic release probability. Together with the increased mEPSC frequency in the TrkC KI CA1 pyramidal cells (Fig. 4E), these fEPSP data suggest that active synapses in TrkC KI mice may possess more glutamate release sites, which is consistent with increased fEPSP slopes, but these release sites exhibit impaired glutamate release probability under action potential-dependent release conditions.

## Aberrant phosphorylation of synaptic proteins caused by loss of TrkC–PTPσ interaction

Next, we addressed the molecular mechanisms underlying the structural and functional phenotypes of TrkC KI synapses. We first searched for synaptic proteins with altered expression by performing unbiased tandem mass tag (TMT) label-based quantitative proteomic assays using synaptosomal samples from the hippocampus and the cortex of TrkC KI and WT littermates but found no differential expression of any proteins, including TrkC (*Ntrk3*), PTPσ (*Ptprs*), VGLUT1 (*Slc17a7*), and PSD-95 (*Dlg4*) (Fig. 5A,B). Combined with the observation of unchanged TrkC and PTPσ expression in hippocampal synaptosomes in western blot experiments (Fig. 1F,G), this suggests that the synaptic phenotypes in TrkC KI mice are unlikely due to altered synaptic expression of TrkC and PTPσ or any other known molecules involved in synaptic organization and function.

TrkC and PTPσ are a receptor-type protein tyrosine kinase and phosphatase, respectively, involved in several protein phosphorylation signaling pathways (Barbacid, 1994; Chagnon et al, 2004; Cornejo et al, 2021; Huang and Reichardt, 2003; Naito et al, 2016; Takahashi and Craig, 2013; Tonks, 2006; Um and Ko, 2013). Thus, we hypothesized that the synaptic phenotypes caused by loss of TrkC–PTPσ interaction in TrkC KI mice are due to changes in the phosphorylation status of synaptic proteins. To test this and to better understand the intracellular mechanisms involved in the TrkC–PTPσ interaction, we next performed unbiased TMT label-based quantitative phosphoproteomic assays using total tissue lysates from the hippocampus and cortex of TrkC KI and WT mice.

In the hippocampus, 223 hyperphosphorylated and 27 hypophosphorylated peptides were identified (Fig. 5C), and 98.0% of them (245 out of 250) met a confidence condition defined by a cutoff curve based on a false discovery rate (FDR) of 0.1 with s0 = 0.1 in the Perseus software (Tyanova et al, 2016). In the cortex, 264 hyperphosphorylated and 740 hypophosphorylated peptides were isolated (Fig. 5D), and 98.4% of them (988 out of 1004) met a confidence condition defined by a cutoff curve based on an FDR of 0.05 with s0 = 0.1. To reduce the inclusion of false positives, we looked for peptides with differential phosphorylation in both the hippocampus and the cortex and found 51 common peptides (46 hyperphosphorylated and 5 hypophosphorylated), belonging to a total of 48 proteins (Fig. 5E,F). Several of the isolated proteins have well-known synaptic functions: Shank3, an excitatory postsynaptic scaffold protein (Monteiro and Feng, 2017); Piccolo (*Pclo*), an active zone protein (Garner et al, 2000; Gundelfinger et al, 2015); dynamin-1 (*Dnm1*), a regulator of clathrin-mediated endocytosis (Ferguson and De Camilli, 2012); Caps-1 (*Cadps*) and rabphilin 3A (*Rph3a*), regulators of vesicle exocytosis (Jockusch et al, 2007; Li et al, 2022); ankyrin-2 (*Ank2*) and spectrin beta, non-erythrocytic 1 (*Sptbn1*), cytoskeletal binding proteins that play multiple functions including in presynaptic organization (Fu et al, 2015; Smith and Penzes, 2018); and GluN2b (*Grin2b*), an NMDA-type glutamate receptor subunit. Because suitable antibodies to validate changes at the identified phosphorylation sites by western blotting are unavailable, we performed Phos-Tag gel electrophoresis (Kinoshita et al, 2006) to assess the overall phosphorylation level of ITPKA, which had one of the highest hyperphosphorylation fold changes and for which many isolated peptides were hyperphosphorylated in the TrkC KI hippocampus (Appendix Fig. S6A). These experiments validated the hyperphosphorylation of ITPKA in TrkC KI hippocampal brain lysates (Appendix Fig. S6).

To better understand in which biological functions the 48 proteins as a group play a role, we next performed Gene Ontology (GO) enrichment analysis (Ashburner et al, 2000) (Fig. 5G; Appendix Table S1). Interestingly, although we used total tissue lysates rather than synaptosomal lysates in the phosphoproteomic experiments, the top eight significantly enriched GO terms in the "Cellular Components" (CC) and the top five in the "Biological Processes (BP)" categories were related to synapses (Fig. 5G; Appendix Table S1). We therefore further performed SynGO

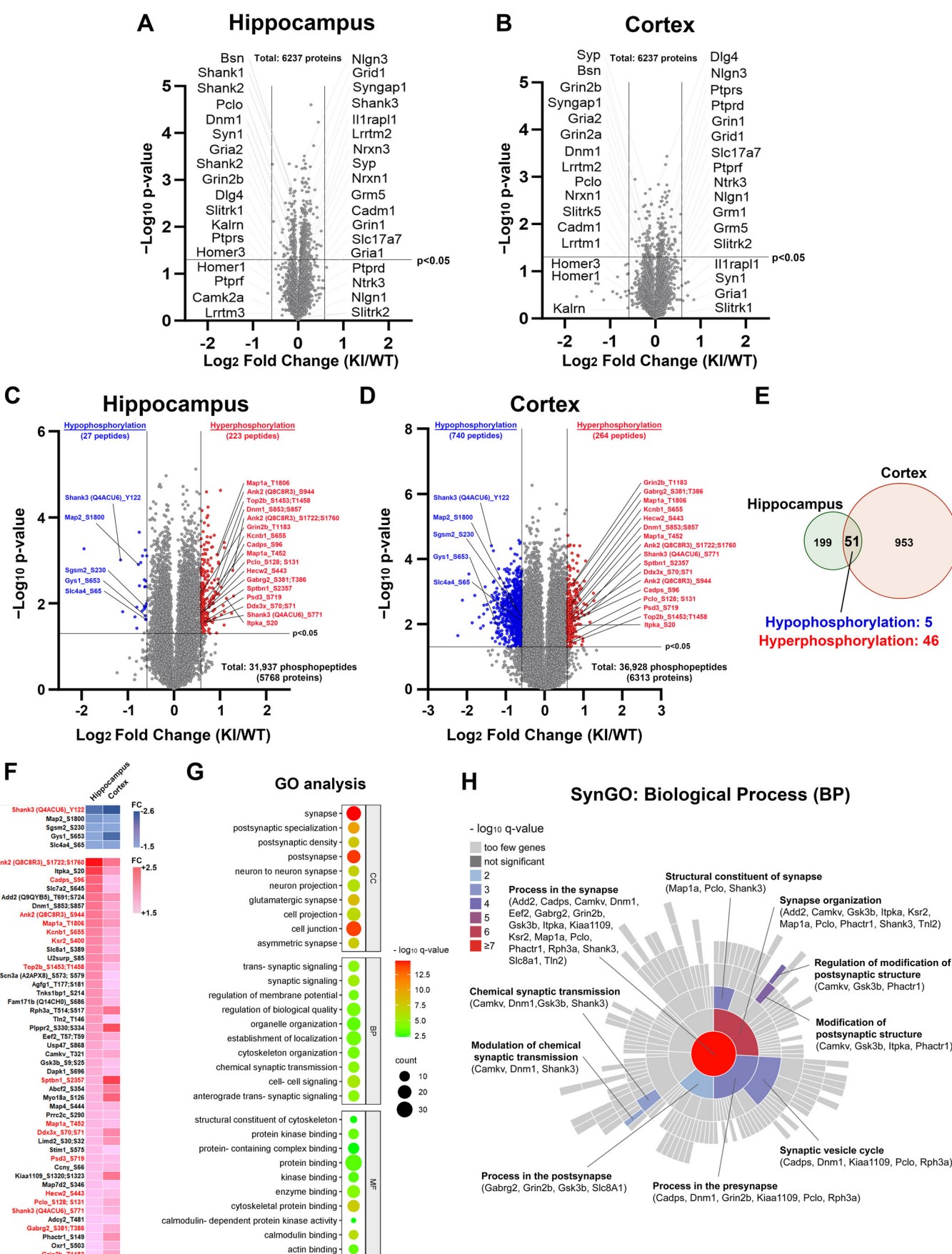

**Figure 5. Unbiased quantitative phosphoproteomics detected altered phosphorylation in a group of synaptic proteins.**

(A, B) Volcano plots showing no proteins are differentially expressed in hippocampal (A) or cortical (B) synaptosomes from TrkC KI mice and WT littermates. Differential protein expression was determined by a criteria of absolute fold change ($|FC|$) greater than 1.5 and $P$ value in Student's $t$ test less than 0.05. $n = 5$ and three mice per genotype for the hippocampus and the cortex, respectively. (C, D) Volcano plots showing altered phosphorylated peptides in the hippocampus (C) and the cortex (D) of TrkC KI mice and WT littermates. Differential phosphopeptide expression was determined by a criteria of $|FC|$ greater than 1.5 and $P$ value in Student's $t$ test less than 0.05. $n = 4$ mice per genotype for the hippocampus and the cortex. (E) Venn diagram showing the number of peptides with altered phosphorylation in the hippocampus and the cortex of TrkC KI mice. (F) Heatmap showing the fold change (FC) of the altered phosphorylation sites detected in both the hippocampus and the cortex. Molecules indicated in red are listed in the SFARI Human Gene database. (G) GO analysis of the phosphorylated molecules altered in both the hippocampus and the cortex in TrkC KI mice for Cellular Component (CC), Biological Process (BP) and Molecular Function (MF). Enrichment for the GO term was determined with g:Profiler (Fisher's Exact test with multiple testing correction based on the g:SCS algorithm). (H) SynGO analysis of the phosphorylated molecules altered in both the hippocampus and the cortex in TrkC KI mice for Biological Process. 18 of the altered 48 genes have Biological Processes annotation in the SynGO analysis. 10 of 208 Biological Processes terms were significantly enriched at 1% FDR (testing terms with at least three matching input genes).

enrichment analysis (Koopmans et al, 2019) (Fig. 5H) and found that many of the 48 isolated proteins were enriched for terms corresponding to biological processes of synapses, such as "synaptic vesicle cycle", "modification of postsynaptic structure" and "modulation of chemical synaptic transmission". These terms are highly relevant to the experimentally observed synaptic phenotypes in TrkC KI mice, including the alterations of SV clustering, PSD structure, basal synaptic transmission and glutamate release probability (Figs. 2–4). These bioinformatic results support that the TrkC–PTPσ complex is involved in excitatory synapse organization and function through regulating the phosphorylation of a particular group of key synaptic proteins.

Curiously, the above GO analysis and SynGO analysis using the common phosphorylated molecules in both the hippocampus and the cortex did not detect terms related to neurotrophin signaling such as "neurotrophin TRK receptor signaling pathway" (GO:0048011), "neurotrophin signaling pathway" (GO:0038179), or the other neurotrophin-related SynGO term (GO:0099183) as significantly enriched (Appendix Table S1). To eliminate the possibility that this could be a false negative because we used a very stringent analytical condition to reduce possible false positives, we performed the analysis without requiring differential phosphorylation in both tissues. However, there was still no enrichment for the above neurotrophin-related GO or SynGO terms (Appendix Table S2 and Appendix Fig. S7). Instead, the terms detected by the additional SynGO analysis fell into categories related to synapses: "synapse organization", "synaptic vesicle cycle" and "chemical synaptic transmission" (Appendix Fig. S7). We further performed Kyoto Encyclopedia of Genes and Genomes (KEGG) analysis to assess relevant signaling pathways (Kanehisa and Goto, 2000) (Appendix Table S2). Again, neither the hippocampal nor cortical molecules were significantly enriched for the term "Neurotrophin signaling pathway (KEGG:04722)" or other terms relevant to canonical signaling pathways of the Trk family such as "PI3K-Akt signaling pathway (KEGG:04151)" or "MAPK signaling pathway (KEGG:04010)" (Appendix Table S2). The volcano plots for the hippocampal and the cortical phosphoproteomic data showed no alteration in the phosphorylation of Mapk1/Erk2, Mapk3/Erk1, Akt1, Akt2, Akt3, or PLC-γ (Plcg1) in TrkC KI mice (Appendix Fig. S8). Our western blot experiments further validated that there was no change in the phosphorylation of Mapk1/3, Akt or PLC-γ (Appendix Fig. S8). Given that dorsal root ganglion (DRG) development depends on NT-3 signaling (Klein et al, 1994), we further examined DRGs and found normal development of DRG neurons and normal afferent projections of them into the

spinal cord in TrkC KI mice (Appendix Fig. S9). Together, these results suggest that loss of TrkC–PTPσ interaction has no significant effect on canonical Trk signaling pathways.

Of note, 15 of the 48 proteins with differential phosphorylation in both the hippocampus and the cortex are listed in the SFARI gene database (Abrahams et al, 2013) as genes associated with neurodevelopmental disorders including ASDs, intellectual disability (ID), attention deficit hyperactivity disorders (ADHD) and/or bipolar disorder (Fig. 5F, marked in red and Appendix Table S3). More interestingly, we performed Human Phenotype Ontology (HPO) analysis and found that the 48 proteins were enriched for "Autistic behavior (HP:0000729) ($q$ value = 0.049, Appendix Table S4)". Given previous human genetic studies show that the TrkC gene *NTRK3* and the PTPσ gene *PTPRS* are associated with neuropsychiatric disorders such as panic disorder, OCD, schizophrenia, and ASDs (Alonso et al, 2008; Armengol et al, 2002; Gratacos et al, 2001; Muinos-Gimeno et al, 2009; Otnaess et al, 2009; Takahashi and Craig, 2013; Um and Ko, 2013; Verma et al, 2008), our results suggest that loss of TrkC–PTPσ interaction in vivo may result in neurodevelopmental disorder- and/or neuropsychiatric disorder-like behaviors.

## Impairment of social novelty in TrkC KI mice

We next performed behavioral assessments of TrkC KI mice. Given that a major behavioral abnormality in ASDs is social defects (Lai et al, 2014), we carried out a three-chamber sociability and social novelty test. TrkC KI and WT mice of both sexes performed similarly in the sociability assessment, preferring to explore the cage containing a stranger mouse to one containing an inanimate object and staying in contact with the stranger mouse for a similar amount of time (Fig. 6A), suggesting that TrkC KI mice do not show altered sociability or social anxiety. In contrast, in the social novelty assessment phase, whereas WT mice showed a preference for exploring the cage containing a novel stranger mouse, TrkC KI mice spent an equivalent amount of time in the familiar and novel mouse cages (Fig. 6A). Thus, TrkC KI mice exhibited no preference for social novelty, suggesting an impairment of social learning and/ or memory in the TrkC KI mice.

To assess other types of learning and memory in TrkC KI mice, we performed a Y-maze spontaneous alternation test and found that there was no significant difference in the percentage of arm alternation (Fig. 6B), suggesting the TrkC KI mice have normal spatial working memory. We further performed contextual and cued fear conditioning tests to evaluate fear learning and memory

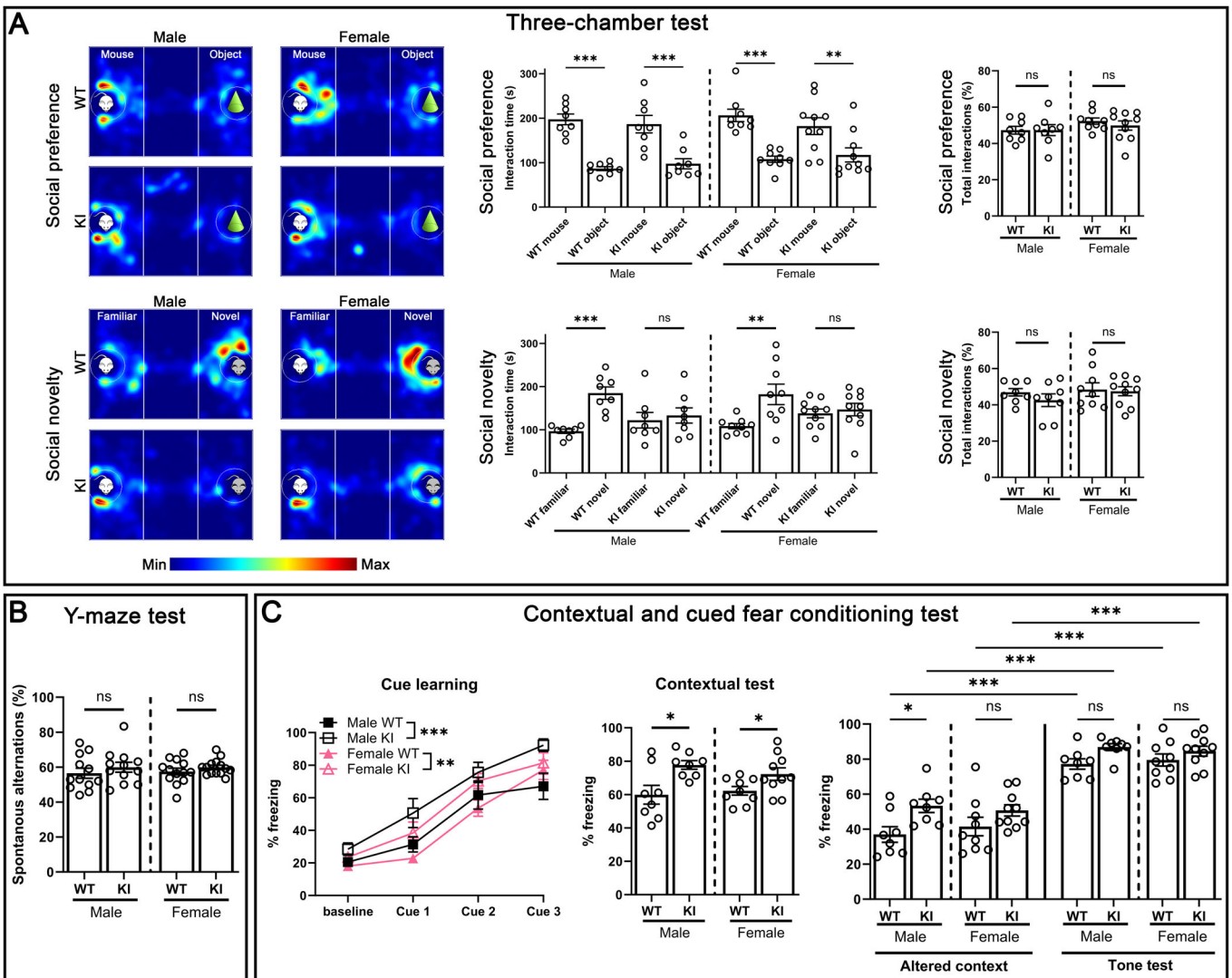

**Figure 6.  TrkC KI mice show impaired social novelty.**

(A) Three-chamber test showing normal social preference behavior in TrkC KI mice (upper) and social novelty test showing significantly less preference to a novel mouse than a familiar one in TrkC KI mice of both sexes (lower). Thus, TrkC KI mice show impaired social novelty but normal sociability. Equivalent total contact time with stranger mice in the three-chamber tests, suggesting no social anxiety in the TrkC KI mice (upper/lower right). One-way ANOVA with post hoc Šídák's multiple comparison test, $F_{(7,62)} = 11.91$, $P = 1.6 \times 10^{-9}$ for social preference, ***$P$ (WT) $= 7.6 \times 10^{-6}$ and ***$P$ (KI) $= 0.00032$ for males, and ***$P$ (WT) $= 2.3 \times 10^{-5}$ and **$P$ (KI) $= 0.0043$ for females. One-way ANOVA with post hoc Šídák's multiple comparison test, $F_{(7,62)} = 4.397$, $P = 0.00051$ for social novelty, ***$P$ (WT) $= 0.00065$ and $P$ (KI) $= 0.98$ for males, and **$P$ (WT) $= 0.0029$ and $P$ (KI) $= 0.99$ for females. Student's $t$ tests for interaction times, $P$ (male) $= 0.99$ and $P$ (female) $= 0.45$ for social preference, and $P$ (male) $= 0.30$ and $P$ (female) $= 0.83$ for social novelty. (B) Y-maze spontaneous alternation test showing no difference between TrkC KI and WT mice of either sex in spontaneous alternation percentage. Thus, TrkC KI mice exhibit normal spatial working memory. Student's $t$ tests, $P$ (male) $= 0.40$ and $P$ (female) $= 0.29$. (C) Fear conditioning test. The fear acquisition curve (left) made by the percentage of freezing responses shows that TrkC KI mice as well as WT mice in both sexes have the ability to acquire cued fear. Two-way repeated measures ANOVA, $F_{(1,56)} = 13.89$, ***$P = 0.00045$ for males and $F_{(1,68)} = 9.632$, **$P = 0.003$ for females. Quantification of the percentage of freezing responses in the contextual test (middle). Student $t$ tests, *$P$ (male) $= 0.013$ and *$P$ (female) $= 0.045$. Quantification of freezing responses in the altered-context test (context discrimination) and the tone test (cued fear memory test) (right). Two-way ANOVA with Šídák's multiple comparisons test, $F_{(1, 28)} = 11.54$, **$P = 0.0021$ in genotype and $F_{(1, 28)} = 86.43$, ***$P = 4.7 \times 10^{-10}$ in test for males, and $F_{(1, 34)} = 5.301$, *$P = 0.028$ in genotype and $F_{(1, 34)} = 109.8$, ***$P = 3.5 \times 10^{-12}$ in test for females. In genotype comparison, *$P$ (male) $= 0.018$ and $P$ (female) $= 0.32$ in the altered-context test, and $P$ (male) $= 0.57$ and $P$ (female) $= 0.72$ in the tone test. In comparison between the two tests, ***$P$ (WT male) $= 2.7 \times 10^{-7}$, ***$P$ (KI male) $= 2.3 \times 10^{-5}$, ***$P$ (WT female) $= 6.0 \times 10^{-8}$, and ***$P$ (KI female) $= 1.1 \times 10^{-7}$. TrkC KI mice of both sexes show normal contextual fear memory, contextual discrimination and cued fear memory. $n = 12$ WT male, 12 KI male, 13 WT female and 14 KI female littermate mice for Y-maze, $n = 8$ WT males, 8 KI males, 9 WT females and 10 KI females for the other tests. Data are presented as mean ± SEM. ns not significant. Source data are available online for this figure.

of environmental cues associated with adverse experiences. TrkC KI mice of both sexes showed equivalent or enhanced freezing responses relative to WT littermates in the conditioning session on day 1, in the contextual session on day 2, and in the cue session on day 3, suggesting that TrkC KI mice of both sexes do not have impaired fear acquisition, contextual fear memory, contextual discrimination or cued fear memory (Fig. 6C).

## Enhanced anxiety-related avoidance behaviors in TrkC KI mice

In the fear conditioning test, we observed an interesting tendency for TrkC KI mice to display enhanced freezing responses compared to WT mice (Fig. 6C). Furthermore, when we attempted to perform object recognition tests for memory assessment, the TrkC KI mice spent most of the test sessions in the corners of the open field and avoided all of the objects, making it impossible to conduct such experiments. The tendency towards enhanced freezing and reduced object contact could be due to increased anxiety and/or avoidance reactions to something unpleasant and/or unfamiliar. Moreover, while the general observations of TrkC KI mice did not reveal obvious differences in mouse condition or transfer behavior (Appendix Fig. S10), during handling, the TrkC KI mice tended to try to escape by jumping away from the experimenter (Fig. 7A), which was not observed at all with their WT littermates. Given several previous human genetic studies showing linkage of *NTRK3* with anxiety-related disorders including panic disorder and OCD, and mood disorders including major depression (Alonso et al, 2008; Armengol et al, 2002; Gratacos et al, 2001; Muinos-Gimeno et al, 2009; Takahashi and Craig, 2013; Um and Ko, 2013; Verma et al, 2008), we further performed behavioral experiments to assess anxiety, avoidance and depression behaviors in TrkC KI mice.

To test whether TrkC KI mice have an increase in their overall anxiety level, we performed two standard anxiety-related tests: open-field tests and elevated plus maze (EPM). In the open-field test, there was no significant difference in the time spent in the anxiogenic center arena between TrkC KI mice and WT mice of either sex (Fig. 7B). TrkC KI females did however travel less than WT females (Fig. 7B), suggesting reduced exploration due to an anxiogenic brightly illuminated and broad open space. This phenotype was not due to a locomotion defect as they exhibited equivalent velocity during exploration of the arena (Fig. 7B), and this phenotype of reduced distance traveled was not observed in other behavioral tests. In the EPM test, TrkC KI and WT mice of both sexes entered the open arms with comparable frequency and spent equivalent time in the open arms (Fig. 7C). These results suggest that TrkC KI mice do not appear to have an increase in overall anxiety level, rather they may display reduced exploration behavior in an anxiogenic open space.

We additionally performed a marble burying test to assess anxiety and repetitive, compulsive-like behaviors. TrkC KI mice of both sexes buried significantly fewer marbles (~5 marbles) than WT mice (~10 marbles) (Fig. 7D). Similar to our observation while attempting to do the object recognition tests, we observed that TrkC KI mice tended to avoid contact with the marbles, which remained mostly untouched and still at their initial position by the end of the 30-min test, as shown in the representative images (Fig. 7D). To investigate whether TrkC KI mice show repetitive, compulsive-like behavior, we assessed grooming behavior induced by splashing sucrose solution and found that TrkC KI mice of both sexes showed grooming duration and frequency comparable to that of WT mice (Fig. 7E), confirming that the TrkC KI mice do not display repetitive, compulsive-like behavior. Grooming assessments are often used to identify depression-like behavior, as depressed mice exhibit reduced grooming in splash tests or deterioration of coat state (Bouguiyoud et al, 2021; Kalueff et al, 2016; Planchez et al, 2019). In addition to our splash test showing comparable grooming (Fig. 7E), the fur coat state of the TrkC KI mice throughout testing was indistinguishable from that of their WT littermates, as represented in Fig. 1H and Appendix Fig. S10, suggesting that the TrkC KI mice do not exhibit depression-like behaviors.

Lastly, we performed another type of anxiety-related behavioral test, a light and dark transition test in which mice are initially placed in a dark compartment and can enter a lit compartment. TrkC KI mice of both sexes took significantly longer than WT littermates to enter the brightly illuminated compartment but then spent an equivalent amount of time in the lit compartment as their WT littermates (Fig. 7F). Together, our data suggest that TrkC KI mice exhibit enhanced avoidance of unfamiliar, anxiogenic conditions, which more closely resembles panic disorder and/or agoraphobia, rather than an increase in their overall anxiety level as would occur in a generalized anxiety disorder, or an increase in anxiety to social interaction as in a social anxiety disorder.

## Discussion

In this study, we uncovered unexpected in vivo consequences of disrupting the interaction between TrkC and PTPσ. First, TrkC KI mice show no change in the number of excitatory synapses or the number of SVs per bouton in excitatory synapses despite previous in vitro studies demonstrating that TrkC–PTPσ interaction promotes excitatory synaptic differentiation including SV accumulation (Ammendrup-Johnsen et al, 2015; Coles et al, 2014; Han et al, 2016; Takahashi et al, 2011). Instead, TrkC KI mice show notable synaptic phenotypes: aberrant SV clustering and PSD elongation. In addition, our electrophysiological experiments demonstrated that in TrkC KI mice, silent synapses are increased whereas active synapses are decreased. Furthermore, the active synapses are aberrant with abnormally enhanced spontaneous and evoked synaptic transmission, presumably due to increased glutamate release sites, and reduced glutamate release probability. These results suggest that the maturation, but not the formation, of excitatory synapses is impaired in TrkC KI mice. Second, through unbiased quantitative proteomic and phosphoproteomic studies, we showed that loss of TrkC–PTPσ interaction significantly alters the phosphorylation status of multiple synaptic proteins that regulate SV clustering, postsynaptic structure and/or synaptic transmission without affecting canonical neurotrophin receptor signaling pathways. Interestingly, many of these molecules are highly linked with neuropsychiatric disorders. In line with this, our behavioral tests revealed that TrkC KI mice display specific behavioral abnormalities, such as social novelty impairment and heightened avoidance of unfamiliar, anxiogenic conditions. Taken together, we propose that the TrkC–PTPσ synaptic organizing complex plays a role in the maturation of excitatory synapses in vivo, but not in synaptogenesis itself, through regulating synaptic protein

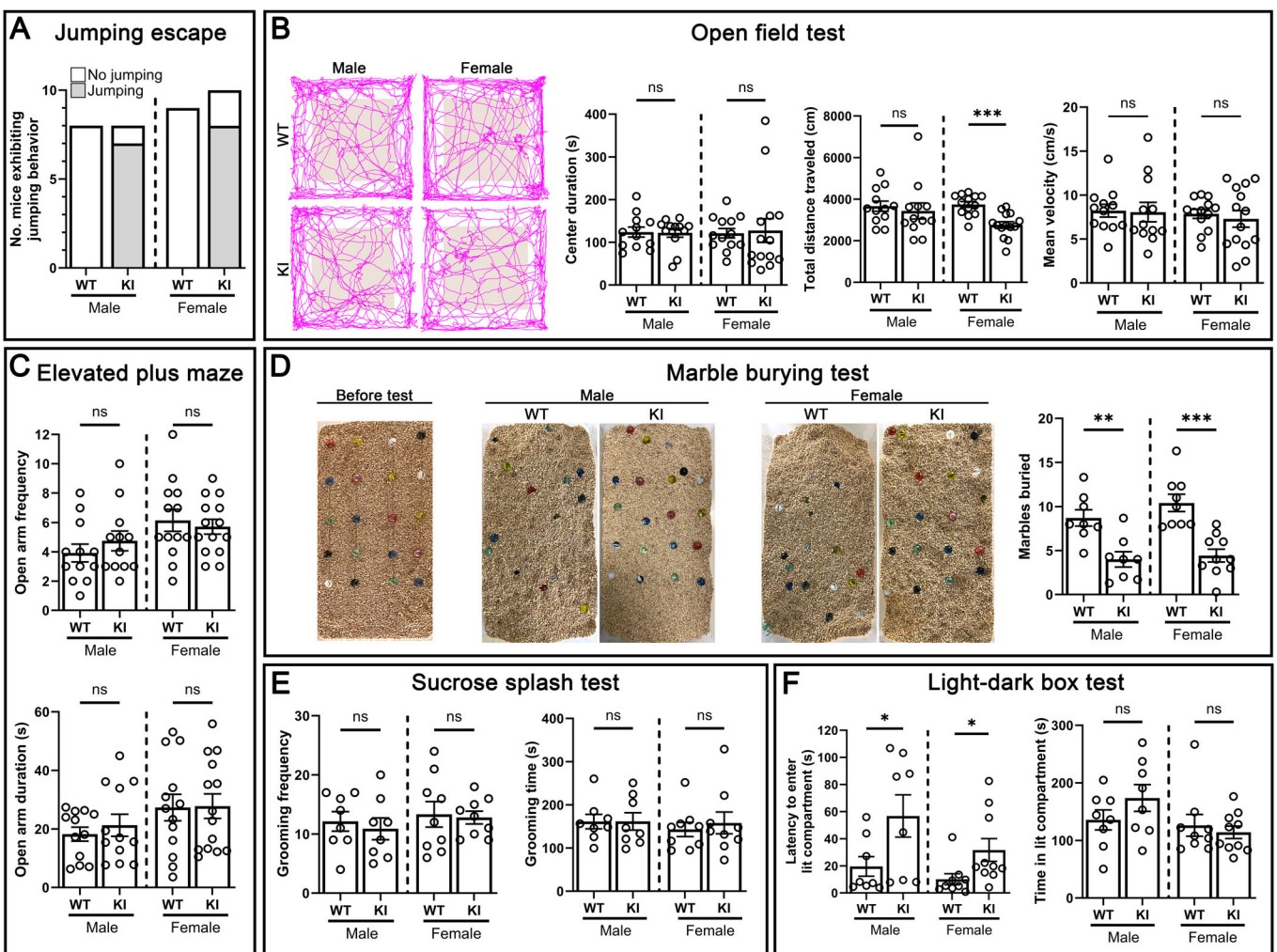

**Figure 7. TrkC KI mice show enhanced avoidance of anxiogenic conditions.**

(A) Number of mice exhibiting jumping behavior in the habituation session. (B) Open-field test showing normal anxiety behavior in TrkC KI mice. There was no difference between TrkC KI mice and WT mice of either sex in center area duration (left graph). Student's *t* tests, *P* (male) = 0.91 and *P* (female) = 0.84. TrkC KI females traveled significantly less than WT females, while TrkC KI males showed equivalent distance to WT males (middle). Student's *t* tests, *P* (male) = 0.62 and ****P* (female) = 9.8 × 10⁻⁵. TrkC KI females as well as TrkC KI males had equivalent velocity compared to their WT littermates (right). Student's *t* tests, *P* (male) = 0.90 and *P* (female) = 0.61. (C) Elevated plus maze test showing normal anxiety behavior in TrkC KI mice. There was no difference between TrkC KI mice and WT mice of either sex in open-arm entry frequency (upper) or duration (lower). Student's *t* tests, *P* (male) = 0.37 and *P* (female) = 0.63 in open-arm frequency and *P* (male) = 0.50 and *P* (female) = 0.94 in open-arm duration. (D) TrkC KI mice buried fewer marbles than WT mice. Many marbles also seem untouched by the TrkC KI mice and remain located in their initial starting position (left image). Student's *t* tests, ***P* (male) = 0.002 and ****P* (female) = 0.00027. (E) Normal grooming behavior in TrkC KI mice after splashing sucrose solution. There was no difference between TrkC KI and WT mice of either sex in grooming frequency (left) or grooming duration (right). Student's *t* tests, *P* (male) = 0.62 and *P* (female) = 0.82 in grooming frequency and *P* (male) = 0.99 and *P* (female) = 0.62 in grooming duration. (F) TrkC KI mice took longer than WT mice to enter the lit compartment (left) but then spent equivalent time in the lit compartment (right). Student's *t* tests, **P* (male) = 0.048 and **P* (female) = 0.040 in latency to enter lit compartment, and *P* (male) = 0.21 and *P* (female) = 0.57 in total duration in lit compartment. *n* = 12 WT male, 12 KI male, 13 WT female and 14 KI female littermate mice for open-field and elevated plus maze tests, *n* = 8 WT males, 8 KI males, 9 WT females and 10 KI females for the other tests. Data are presented as mean ± SEM. ns not significant. Source data are available online for this figure.

phosphorylation, demonstrating a non-canonical function of the neurotrophin receptor TrkC as a synaptic organizer that defines excitatory synaptic properties.

Many in vitro studies of synaptogenic molecules, especially gain-of-function studies based on artificial synapse formation assays and neuronal overexpression experiments, have contributed to identifying and characterizing synaptic organizers, including TrkC and PTPσ (Graf et al, 2004; Linhoff et al, 2009; Scheiffele et al, 2000; Takahashi et al, 2011; Takahashi et al, 2012; Tanabe et al, 2017). On

the other hand, many in vivo loss-of-function studies of these synaptic organizers using mutant mice have shown that the demonstration of in vitro synaptogenic activity does not necessarily indicate that their endogenous function contributes to synapse formation (Dhume et al, 2022; Sudhof, 2017; Varoqueaux et al, 2006). For instance, in the hippocampal CA1 region, VGLUT1 signal is unchanged in mice with global knockout (KO) of neuroligin-1 (Nlgn1) (Blundell et al, 2010), increased in LRRTM1 global KO mice (Linhoff et al, 2009), but decreased in

global double LRRTM1/2 KO mice (Dhume et al, 2022). In addition, it has been challenging to pursue in vivo analysis of the endogenous function of TrkC as a synaptic organizer because it also plays a crucial role as a neurotrophin receptor. Previous studies have generated two different TrkC mutant mouse lines: one in which TrkC tyrosine kinase activity is disrupted but TrkC synaptogenic activity is retained (Klein et al, 1994) and another in which both the synaptogenic and neurotrophin receptor functions of TrkC are disrupted (Tessarollo et al, 1997). However, neither line is suitable for the assessment of endogenous TrkC as a synaptic organizer. Similarly, given that PTPσ has multiple binding partners (Takahashi and Craig, 2013; Um and Ko, 2013), several of which bind to the same domains of PTPσ as TrkC (Takahashi and Craig, 2013; Um and Ko, 2013), it has been challenging to dissect the function of PTPσ with respect to TrkC binding because PTPσ KO approaches cannot clarify which of its binding partners are responsible for any observed phenotypes. To overcome these limitations, our TrkC KI mutant mouse line selectively abolishes TrkC–PTPσ interaction, thereby diminishing the synaptogenic activity of the complex without interfering with TrkC-NT-3 signaling or PTPσ interaction with its other binding partners. Indeed, our TrkC KI mice did not show any of the synaptic or neuronal phenotypes previously observed in mutant mice lacking TrkC kinase activity, such as reduction of excitatory synapse number and PSD length in hippocampal CA1 st. rad. regions (Martinez et al, 1998) or impaired development of DRG neurons (Klein et al, 1994). Thus, this unique genetic strategy provided us with practical advantages for proper assessment of the role of the TrkC–PTPσ synaptic organizing complex in vivo.

Through analysis of the TrkC KI mice, we revealed that TrkC–PTPσ interaction is dispensable for synaptogenesis as synapse numbers remained unaffected. This could be due to compensation by other synaptic organizers, with the high redundancy of synaptic hub networks based on type II RPTPs ensuring mechanistic robustness of synapse formation. In line with this, a previous study showed that knockout of PTPσ does not alter synaptogenesis in cultured neurons or acute brain slices (Sclip and Sudhof, 2020). On the other hand, TrkC KI synapses did show unique structural and functional presynaptic phenotypes: enhanced SV clustering, increased release sites, and diminished glutamate release probability, presumably because TrkC binds no presynaptic organizers other than PTPσ (Naito et al, 2016; Takahashi et al, 2011; Takahashi and Craig, 2013; Um and Ko, 2013). Many different types of presynaptic molecules and their phosphorylation are known to underlie the regulation of SV clustering and SV exo-/ endocytotic recycling for neurotransmitter release, such as synapsin I, an SV-associated protein (Bonanomi et al, 2005; Shupliakov et al, 2011), bassoon and piccolo, active zone proteins (Garner et al, 2000; Gundelfinger et al, 2015), the SNARE protein complex, intracellular membrane fusion machinery (Sudhof and Rothman, 2009; Turner et al, 1999), Caps-1 (Cadps), a synaptic vesicle priming protein (Jockusch et al, 2007; Nojiri et al, 2009), and dynamin-1, a membrane severing protein involved in clathrin-dependent endocytosis (Ferguson and De Camilli, 2012; Imoto et al, 2024; Tomizawa et al, 2003). Importantly, our phosphoproteomic analyses have shown that phosphorylation of several of these molecules and/or their regulators including piccolo, Caps-1 (Cadps), dynamin-1 and rabphilin 3A is altered in TrkC KI mice. Thus, loss of TrkC–PTPσ interaction likely causes abnormalities in

SV clustering and recycling thereby impairing glutamate release through dysregulating the phosphorylation of specific presynaptic molecules, although further studies are necessary to address which of the identified phosphorylation alterations cause the presynaptic phenotypes in TrkC KI mice. Taken together, our data suggest that while dispensable for synapse formation, TrkC–PTPσ interaction governs unique synapse functions that cannot be compensated for by other synaptic organizing molecules through regulating synaptic protein phosphorylation.

In addition to structural abnormalities, we further found that TrkC KI mouse neurons show electrophysiological markers of deficient synapse maturation. In particular, TrkC KI mice have an increased fraction of silent synapses. Throughout early development, the number of silent synapses typically decreases as synapses recruit functional AMPARs in response to ongoing neuronal activity (Isaac et al, 1995; Liao et al, 1995). The excess of silent synapses in TrkC KI mice may reflect a failure in this process, consistent with findings in several rodent models for neurodevelopmental disorders (Wan et al, 2011; Wegener et al, 2018). In addition, assessing mEPSC frequency and I/O responses revealed that spontaneous and evoked basal glutamatergic transmission is enhanced in TrkC KI mice. These results could reflect functional compensation for delayed synapse maturation. Furthermore, the enhanced PPF at short inter-stimulus intervals is consistent with the disorganization of presynaptic terminals and subsequent alterations in the kinetics of vesicular release. Accordingly, aberrant SV clustering could increase glutamate release which would be amplified when two impulses are applied in rapid succession, consistent with increased PPF only being detected at the shortest inter-stimulus intervals. Further testing of presynaptic function will be required to identify the precise mechanisms contributing to these physiological changes. Taken together, our results showing increased vesicle clustering along with increased silent synapses and altered glutamatergic transmission support a model in which loss of TrkC–PTPσ interaction impairs synapse organization and delays synapse maturation, which alters presynaptic function and spurs compensatory upregulation of spontaneous and evoked glutamatergic synaptic transmission at active synapses.

As an interesting postsynaptic phenotype, TrkC KI synapses also show PSD elongation in the st. rad. Our mEPSC experiments show no difference in mEPSC amplitude between TrkC KI and WT mice, suggesting that PSD elongation is not associated with increased postsynaptic AMPARs. Therefore, PSD elongation may be a structural postsynaptic change accompanied by an increase in glutamate release sites at the presynaptic terminals. However, another remaining question is how the loss of TrkC–PTPσ interaction leads to PSD elongation. Our phosphoproteomic assays detected altered phosphorylation of Shank3, a PSD scaffold protein (Monteiro and Feng, 2017), as well as CaM kinase-like vesicle-associated protein (CaMKv) and Inositol trisphosphate 3-kinase A (ITPKA), both of which are known to regulate the morphology of dendritic spines (Erneux et al, 2016; Liang et al, 2016). In particular, the hypophosphorylation of Shank3 tyrosine122 (Y122), located in the linker region between the Shank/ProSAP N-terminal (SPN) domain and ankyrin repeats, is interesting because conformational changes of this N-terminal region are involved in regulating the number and morphology of dendritic spines (Salomaa et al, 2021). Moreover, CaMKv regulates dendritic spine morphology in a phosphorylation-dependent manner

(Liang et al, 2016). Further studies may elucidate whether and how the TrkC–PTPσ complex regulates the molecular properties and functions of these postsynaptic regulators.

The mechanism underlying the plethora of changes in phosphorylation in the TrkC KI mice is also unclear. We observed no change in canonical neurotrophin signaling pathways, suggesting that TrkC kinase activity is unlikely to be significantly affected by loss of TrkC–PTPσ interaction. Altered phosphatase activity of PTPσ is another possibility. However, unlike in the TrkC KI mice, presynaptic KO of PTPσ has no effect on basal synaptic transmission or paired-pulse facilitation (Kim et al, 2020), suggesting that loss of TrkC–PTPσ interaction may not cause a loss of function of presynaptic PTPσ. Further future studies are important to determine whether and how TrkC–PTPσ interaction is involved in the activation of PTPσ phosphatase activity as well as TrkC kinase activity to address how the TrkC–PTPσ complex regulates pre- and postsynaptic protein phosphorylation.

Many human and animal genetic studies have linked synaptic pathology causally with neuropsychiatric conditions (Penzes et al, 2013). TrkC is genetically linked with several different types of neuropsychiatric disorders including panic disorder, OCD, major depression and schizophrenia (Alonso et al, 2008; Armengol et al, 2002; Dierssen et al, 2006; Gratacos et al, 2001; Muinos-Gimeno et al, 2009; Otnaess et al, 2009; Takahashi and Craig, 2013; Um and Ko, 2013; Verma et al, 2008). These disorders could be associated with dysfunction and dysregulation of TrkC both as a neurotrophin receptor and/or as a synaptic organizer. Indeed, neurotrophins including NT-3 have long been proposed as susceptibility genes for major depression and schizophrenia, and administration of NT-3 was shown to have an antidepressant effect on some mouse models for depression (Lin and Tsai, 2004; Shirayama et al, 2002). This suggests that TrkC linkage to major depression and schizophrenia is likely due to its canonical neurotrophic function rather than non-canonical synaptic organizer function. Therefore, the absence of depression-like behavior in the TrkC KI mice, along with our observation of no changes in phosphorylation of canonical Trk signaling molecules and pathways, further support our conclusion that loss of TrkC–PTPσ interaction does not change neurotrophin receptor activity of TrkC. Thus, the neuropsychiatric phenotypes observed in TrkC KI mice are most likely due to the loss of the synaptic organizer function of TrkC, which causes the unique synaptic pathologies described above. Furthermore, TrkC KI mice show some specific behavioral abnormalities such as social novelty impairment and enhanced avoidance behaviors, with normal fear learning/memory, social preference, and repetitive behavior. Although several molecules isolated in our phosphoproteomic analysis may suggest a linkage with ASDs, these behavioral phenotypes suggest that TrkC KI mice do not mimic ASDs, rather they model other specific disorders such as panic disorder and/or agoraphobia.

Future studies are needed to follow-up on some aspects of the current study. First, although our EM data suggest that the increased signal of VGLUT1 in the IHC is likely to result from the enhanced clustering of SVs, it remains possible that this might also result from the altered stoichiometry of VGLUT1 per SV. To address this possibility, it is important to conduct a biochemical analysis of highly-purified SVs (Takamori et al, 2006) and/or a super-resolution microscopy imaging of isolated, single SVs (Upmanyu et al, 2022). Second, the EM analysis was performed using samples prepared by chemical fixation, which has been reported to induce artificial structural changes, thereby complicating the assessment of SVs, especially SVs near presynaptic membranes (Korogod et al, 2015). Indeed, in our EM study, some synapses (mostly in TrkC KI mice) showed dark cloudy structures near presynaptic membranes, due to enhanced SV clustering, which made it difficult to precisely assess the number and distribution of docked SVs. Given that TrkC KI synapses may have more glutamate release sites, it will be important to use a rapid high-pressure freezing method (Korogod et al, 2015; Siksou et al, 2007) to obtain better and more accurate images of SVs, including docked SVs. Third, it will be necessary to generate antibodies that recognize the phosphorylation sites identified in the phosphoproteomic experiments to perform independent experiments to validate these alterations. Finally, it remains unknown whether and which phosphorylation alterations are causatively linked with the synaptic and behavioral phenotypes of TrkC KI mice. One way to address this question would be to generate and characterize KI mice expressing phosphodead or phosphomimetic mutants of the isolated proteins.

In conclusion, we uncovered unique in vivo roles of the endogenous TrkC–PTPσ synaptic organizing complex by characterizing TrkC KI mutant mice. The TrkC–PTPσ complex defines SV organization, PSD structure and excitatory synapse function through regulating synaptic protein phosphorylation without affecting canonical neurotrophic signaling pathways. Meanwhile, disrupting TrkC–PTPσ interaction causes a social novelty defect and anxiety-related avoidance behaviors. Given genetic linkages of TrkC with panic disorder and some other neuropsychiatric conditions, we propose "synaptic phospho-pathology" as their underlying pathological mechanism, providing a novel animal model and new therapeutic insights into *NTRK3*-linked neuropsychiatric disorders.

# Methods

**Reagents and tools table**

| Reagent/resource | Reference or source | Identifier or catalog number |
|---|---|---|
| **Experimental models** | | |
| C57BL/6 embryonic stem cell line Bruce-4 cells (*M. musculus*) | Sigma-Aldrich | CMTI-2 |
| *ACTB:FLPe* B6J mice | The Jackson Laboratory | JAX stock # 005703 |
| C57BL/6J mice | The Jackson Laboratory | JAX stock # 000664 |
| TrkC knock-in (KI) D240A;D242A mice | This study | |
| COS-7 cells | ATCC | CRL-1651 |

| Reagent/resource | Reference or source | Identifier or catalog number |
|---|---|---|
| **Recombinant DNA** | | |
| TrkC KI targeting vector | Gene Bridges GmbH/this study | N/A |
| pCAG-EGFP | Addgene | 11150 |
| Mouse Neurotrophin-3/NT-3 cDNA Clones | SinoBiological | MG50223-CM |
| pCAG-NT-3-myc | This study | N/A |
| **Antibodies** | | |
| Mouse anti-PSD-95 | Thermo Fisher Scientific | MA1-045 |
| Guinea pig anti-VGLUT1 | Sigma-Aldrich | AB5905 |
| Rabbit anti-TrkC | Cell Signaling Technology | 3376 |
| Mouse anti-PTPσ | MediMabs | MM-0020-P |
| Rabbit anti-VGLUT1 | Synaptic Systems | 135302 |
| Rabbit anti-Neurotrophin-3 | Abcam | ab53685 |
| Rabbit anti-β-actin | Abcam | ab8227 |
| Mouse anti-β-actin | Abcam | ab8226 |
| Rabbit anti-myc | Cell Signaling Technology | 2272S |
| Mouse anti-HA | Roche | 12CA5 |
| Chicken anti-MAP2 | Abcam | ab5392 |
| Rabbit anti-phospho-p44/42 MAPK (Erk1/2) (Thr202/Tyr204) | Cell Signaling Technology | 9101 |
| Rabbit anti-p44/42 MAPK (Erk1/2) | Cell Signaling Technology | 9102 |
| Rabbit anti-phospho-AKT (Ser473) | Cell Signaling Technology | 4060 |
| Rabbit anti-AKT | Cell Signaling Technology | 9272 |
| Rabbit anti-phospho-PLCγ1 (Ser1248) | Cell Signaling Technology | 8713S |
| Rabbit anti-PLCγ1 | Cell Signaling Technology | 5690T |
| Rabbit anti-ITPKA | Thermo Fisher Scientific | PA5-75226 |
| Mouse anti-β-actin | Sigma-Aldrich | A5441 |
| Rabbit anti-NeuN-Alexa 647 conjugated | Abcam | ab190565 |
| Rabbit anti-NF200 | MilliporeSigma | N4142 |
| Goat anti-TrkB | R&D Systems | AF1494 |
| Goat anti-TrkC | R&D Systems | AF1404 |
| Goat anti-CGRP | Abcam | ab36001 |
| IsolectinB4 stain-Alexa 647 conjugated | Life Technologies | I32450 |
| Donkey anti-mouse IgG Alexa488 | Jackson ImmunoResearch Laboratories | 711-545-151 |
| Donkey anti-guinea pig IgG Alexa488 | Jackson ImmunoResearch Laboratories | 706-545-148 |
| Donkey anti-guinea pig IgG Alexa594 | Jackson ImmunoResearch Laboratories | 706-585-148 |
| Donkey anti-rabbit IgG Alexa488 | Jackson ImmunoResearch Laboratories | 711-545-152 |
| Donkey anti-rabbit IgG Alexa594 | Jackson ImmunoResearch Laboratories | 711-585-152 |
| Donkey anti-chicken IgY AMCA | Jackson ImmunoResearch Laboratories | 703–155–155 |
| Donkey anti-goat IgG Alexa488 | Jackson ImmunoResearch Laboratories | 706-545-147 |
| Donkey anti-mouse IgG HRP | Jackson ImmunoResearch Laboratories | 715-035-151 |
| Donkey anti-rabbit IgG HRP | Jackson ImmunoResearch Laboratories | 711-035-152 |
| **Oligonucleotides and other sequence-based reagents** | | |
| PCR primers | This study | Methods |
| **Chemicals, enzymes, and other reagents** | | |
| Superfrost Plus Micro Slide | VWR | CA48311-703 |
| Eukitt Quick-hardening mounting medium | Sigma-Aldrich | 03989 |

| Reagent/resource | Reference or source | Identifier or catalog number |
|---|---|---|
| Optimal Cutting Temperature compound | Tissue-Tek/VWR | 25608-930 |
| Albumin from bovine serum | Sigma-Aldrich | A9647 |
| Normal Donkey Serum | Jackson ImmunoResearch Laboratories | 017-000-121 |
| Triton X-100 | Sigma-Aldrich | T9284 |
| DAPI | Sigma-Aldrich | D9542 |
| HEPES (for synaptosome and lysate preparations) | Thermo Fisher Scientific | 15630-080 |
| Protease inhibitors | Roche | 5892953001 |
| Phosphatase inhibitors | Roche | 04906837001 |
| DTT (Dithiothreitol) | G-biosciences | BC99 |
| DC Protein Assay Kit | Bio-Rad | #500-0112 |
| Protein G Sepharose 4 Fast Flow | GE HealthCare Life Science | 17-0618-01 |
| Immun-Blot PVDF Membrane | Bio-Rad | 1620177 |
| Clarity™ Western ECL | Bio-Rad | 1705061 |
| Glutaraldehyde | Sigma-Aldrich | G5882 |
| Paraformaldehyde | Electron Microscopy Sciences | 19202 |
| Sodium cacodylate trihydrate | MilliporeSigma | C0250 |
| PhosSTOP | Sigma-Aldrich | 4906837001 |
| TransIT-LT1 | Mirus Bio | MIR2305 |
| Hibernate E | BrainBits | HE500 |
| Poly-L-lysine hydrobromide | Sigma-Aldrich | P2636-1G |
| RIPA buffer | MilliporeSigma | 20-188 |
| Calf-intestinal alkaline phosphatase | Promega | M2825 |
| SuperSep™ Phos-Tag™ pre-cast gels | FUJIFILM | 198-17981 |
| TEMED | Bioshop | TEM001.50 |
| Ammonium persulfate | Sigma-Aldrich | A3678 |
| D-Sucrose | Fisher Bioreagents | BP220-1 |
| D-($+$)-Glucose | Sigma-Aldrich | G8270 |
| Sodium bicarbonate (NaHCO$_3$) | Sigma-Aldrich | S6014 |
| Potassium chloride (KCl) | Sigma-Aldrich | P9333 |
| Sodium phosphate monobasic (NaH$_2$PO$_4$) | BioShop | SPM400 |
| Calcium chloride solution (CaCl$_2$) | BioShop | CCL333 |
| Magnesium sulfate solution (MgSO$_4$) | MilliporeSigma | 83266 |
| Sodium chloride (NaCl) | Sigma-Aldrich | S9888 |
| Cesium gluconate | Hellobio | HB4822 |
| Cesium chloride | Sigma-Aldrich | 289329 |
| Guanosine 5′-triphosphate sodium salt hydrate (Na GTP) | Sigma-Aldrich | 51120 |
| Adenosine 5′-triphosphate magnesium salt (Mg ATP) | Sigma-Aldrich | A9187 |
| Ethylene glycol-bis(2-aminoethylether)-N,N,N′,N′-tetraacetic acid (EGTA) | Sigma-Aldrich | E0396 |
| Phosphocreatine disodium salt hydrate | Sigma-Aldrich | P7936 |
| HEPES (for electrophysiology) | Sigma-Aldrich | H3375 |
| QX-314-Cl | MilliporeSigma | 552233 |
| Tetrodotoxin citrate | StressMarq Biosciences | SIH-603 |
| Bicuculline methiodide | Sigma-Aldrich | 14343 |
| Cesium hydroxide solution | Sigma-Aldrich | 232041 |

| Reagent/resource | Reference or source | Identifier or catalog number |
|---|---|---|
| Guanidine-HCl | FUJIFILM | 071-02891 |
| HEPES (for proteomics) | DOJINDO | 340-01371 |
| Tris (2-carboxyethyl) phosphine (TCEP) | Nacalai Tesque | 07277-16 |
| Chloroacetamide | Sigma-Aldrich | C0267 |
| Methanol | FUJIFILM Wako | 138-14521 |
| Chloroform | Nacalai Tesque | 08402-84 |
| RapiGest SF | Waters | 186001861 |
| Triethylammonium bicarbonate | FUJIFILM | 206-38381 |
| Trypsin/Lys-C mix, Mass Spec Grade | Promega | V5071 |
| TMTpro 16-plex Label Reagent Set | Thermo Scientific | A44520 |
| TMT 10-plex Isobaric Label Reagents and Kit | Thermo Scientific | 90110 |
| Acetonitrile, Super Dehydrated | FUJIFILM | 018-22901 |
| 50% Hydroxylamine Solution | FUJIFILM | 088-07221 |
| High-Select Fe-NTA Phosphopeptide Enrichment Kit | Thermo Scientific | A32992 |
| Ammonium formate | FUJIFILM | 010-03122 |
| Acetonitrile | FUJIFILM | 012-19851 |
| Trifluoroacetic acid | FUJIFILM | 206-10731 |
| **Software** | | |
| GraphPad Prism 10 | https://www.graphpad.com | |
| ImageJ | https://imagej.nih.gov/ij/index.html | |
| Metamorph 7.8 | Molecular Devices | |
| Volocity 6.0 | https://www.volocity4d.com | |
| Pclamp 11.2 | https://www.moleculardevices.com/ | |
| MiniAnalysis 6.0 | https://minianalysis.software.informer.com/6.0/ | |
| WinLTP 3.00 | https://winltp.com/ | |
| EthoVision XT | Noldus | |
| Proteome Discoverer 2.4 | Thermo Scientific | |
| Perseus 2.0.3.1 | https://maxquant.net/perseus/ | |
| **Other** | | |
| CryoStar NX70 | Thermo Scientific | |
| ChemiDoc™ XRS+ Imager | Bio-Rad | |
| DM6 fluorescence microscope with Hamamatsu C11440 ORCA-Flash 4.0 camera | Leica | |
| SP8 confocal microscope | Leica | |
| FEI Tecnai G2 Spirit TEM with Gatan Ultrascan 4000 CCD Camera Model 895 | Tecnai | |
| Multiclamp 700B amplifier | Molecular Devices | |
| Digidata 1550 | Molecular Devices | |
| Stimulation Box | Digitimer Ltd. | Model DS3 |
| Upright microscope | Nikon | Nikon Eclipse FN1 |
| CCD camera | DAGE-MTI | IR-2000 Infrared Video Camera |
| VT1200S vibratome | Leica Biosystems | |
| Incubation chamber | Scientific Systems Design | BSK12 |
| Submerged recording chamber | Warner Instruments | RC-26 |
| Interface chamber | Scientific Systems Design | |
| P-97 microelectrode puller | Sutter Instrument | |

| Reagent/resource | Reference or source | Identifier or catalog number |
|---|---|---|
| pH meter | Fisherbrand | Accumet AB315 |
| Osmometer | ADVANCED INSTRUMENTS | Osmo1 |
| Borosilicate glass electrodes | Sutter Instruments | BF150-86-10 |
| Bipolar nickel-chromium electrode | A-M Systems | 762000 |
| Vanquish DUO UHPLC | Thermo Scientific | VQDUO-DUALLC |
| 4.6 × 250 mm Xbridge BEH130 C18 column with 3.5 μm particles | Waters | 186003943 |
| Savant SpeedVac | Thermo Scientific | SPD1010 |
| EASY-nLC 1200 | Thermo Scientific | LC140 |
| Q Exactive Plus mass spectrometer | Thermo Scientific | IQLAAEGAAPFALGMBDK |
| C18 reversed-phase column 75 μm × 155 mm | Nikkyo Technos | NTCC-360/75-3-155 |
| ProteoSave Vial | AMR Inc | PSVial100 |
| 9 mm PP-Screw Cap | AMR Inc | PSVCapS100 |
| Y-maze | Noldus | |
| Fear Conditioning System | Noldus | |
| Three Chambered Sociability Cage | Noldus | |

## Materials and protocols

### Animal and ethics statement

All animal experiments were carried out in accordance with Canadian Council on Animal Care guidelines and approved by the IRCM Animal Care Committee and the York University Animal Care Committee. Mice were group-housed (two to five per cage) and maintained on a 12-h light/dark cycle. For all experiments and analyses, the experimenters were blind to genotype.

### Generation of TrkC knock-in D240A;D242A mouse line

The TrkC knock-in (KI) targeting vector was generated in collaboration with Gene Bridges GmbH. The 13.4-kb construct contains exon 6 of the *NTRK3* gene, with the D240A D242A mutations, adjacent to two FRT sites flanking a NeoR/KanR cassette. C57BL/6 embryonic stem cell line Bruce-4 cells (RRID: CVCL_K037) were electroporated with the targeting vector and selected using G148. To confirm homologous recombination, DNA from resistant cells was digested with EcoRV and screened by Southern blotting using an external 5′ probe, and separately digested with StuI for screening with an external 3′ probe. Confirmed cells were used to generate chimeric mice from C57BL/6J mice, which were further screened by Southern blotting using the external 5′ probe. The chimeric mice were crossed with C57/BL6J mice and confirmed for correct homologous recombination using the 5′ and 3′ probes, as well as a NeoR cassette probe. Genotyping by PCR used the following primers: NTRK3 wild-type 5′ primer 5′-GAGTGAGAGCTGTCAATCAGC-3′, NTRK3 exon 6 3′ primer 5′-AGAGCCATTGCAAGTGATCACGGCATTG-3′ and FRT site 3′ primer 5′-TCGCCTTCTATCGCCTTCTTGACGAGTTC-3′. Expected PCR product sizes are 592 bp (WT) and 533 bp (KI). Following this, the NeoR/KanR cassette was removed by crossing the heterozygous KI mice with *ACTB:FLPe* B6J mice (Rodriguez et al, 2000) (The Jackson Laboratory, JAX stock # 005703, *B6.Cg-Tg(ACTFLPe)9205Dym/J*) and removal of the cassette was confirmed by the absence of a band when genotyping with NTRK3 wild-type 5′

and FRT site 3′ primers. Further PCR validation, as well as all subsequent genotyping, was carried out using the NTRK3 wild-type 5′ and NTRK3 exon 6 3′ primers, which produce 592 bp (WT) or 683 bp (KI) bands. TrkC KI heterozygous mice lacking the NeoR cassette were then backcrossed with C57BL/6J mice to obtain TrkC KI heterozygous mice that no longer carried the *ACTFLPe* transgene. Finally, before experimentation and analysis, the mice were backcrossed with C57BL/6J mice more than six times for immunohistochemistry and synaptosome western blotting experiments and more than 9 times for all other experiments. At each generation, the transmission of the point mutations was confirmed through sequencing analysis using a 5′ primer 5′-AGCTCCTCTGTGGACTGTCACCTG-3′ and a 3′ primer 5′-TCACCAGGGTCAAGTTGATGGCATGTAC-3′.

### Nissl staining

Four-week-old WT and homozygous KI littermates were anesthetized with a ketamine/xylazine mixture (10 mg/mL ketamine, 1 mg/mL xylazine, in 0.9% saline; dose: 0.1 mL/g body weight) and perfused transcardially with phosphate-buffered saline (PBS, pH 7.4) followed by 4% paraformaldehyde in PBS. The brain was removed and post-fixed for 2 h at 4 °C. After embedding the brains in paraffin, 5-μm sagittal or coronal sections were cut using a microtome and mounted on Superfrost Plus slides (VWR). Nissl staining was performed on deparaffinized sections by immersion into warm 0.1% cresyl violet solution for 10 min, rinsing three times in distilled water and differentiating in 95% ethanol. Slides were then dehydrated in 100% ethanol, cleared in xylene and mounted with Eukitt® Quick-hardening mounting medium (Sigma-Aldrich).

### Immunohistochemistry

Four-week-old TrkC homozygous KI mice (TrkC KI mice) and WT littermate mice were anesthetized with a ketamine/xylazine mixture, followed by transcardial perfusion with PBS for blood removal. The brains were then extracted, frozen in Optimal Cutting

Temperature (O.C.T.) compound (Tissue-Tek) and stored at −80 °C until sectioning. Coronal brain sections (14-µm thick) were cut using a Cryostat (Thermo Scientific, CryoStar NX70) and mounted on Superfrost Plus slides (VWR). Slides were submerged in a −20 °C methanol bath for 15 min, washed three times with PBS for 10 min each, then blocked using blocking solution (PBS containing 5% bovine serum albumin (BSA) (wt/vol), 2.5% normal donkey serum (vol/vol)) and 0.25% Triton X-100 (vol/vol)). In double immunolabeling for VGLUT1 and PSD-95, same sections were incubated overnight at 4 °C with anti-VGLUT1 (1:2000, polyclonal guinea pig, AB5905, Sigma-Aldrich) and anti-PSD-95 (1:500, monoclonal mouse, MA1-045, clone 6G6-1C9, Thermo Fisher Scientific) at once. For TrkC immunolabeling, anti-TrkC (1:500, monoclonal rabbit, 3376, clone C44H5, Cell Signaling Technology) was used. Following primary antibody incubation, slides were washed with PBS three times for 10 min each, then sections were incubated for 1 h at room temperature with a mixture composed of DAPI (100 ng ml$^{-1}$) and the following secondary antibodies: Alexa594-conjugated donkey anti-guinea pig IgG (1:500, 706-585-148, Jackson ImmunoResearch) and Alexa488-conjugated donkey anti-mouse IgG (1:500, 711-545-151, Jackson ImmunoResearch) in double immunolabeling for VGLUT1 and PSD-95, and Alexa488-conjugated donkey anti-rabbit IgG (1:500, 711-545-152, Jackson ImmunoResearch) in TrkC immunolabeling. Slides were then washed with PBS three times for 10 min each, and sections were sealed with elvanol mounting reagent (Tris-HCl, glycerol, and polyvinyl alcohol with 2% 1,4-diazabi-cyclo[2,2,2]octane).

### Quantitative fluorescent image analysis

All imaging and image analysis were done while blind to the experimental conditions. All the data for imaging were collected in random order. Analysis was performed using Metamorph 7.8 (Molecular Devices). For immunohistochemistry analysis, quantification of VGLUT1 and PSD-95 puncta intensity in the stratum radiatum and stratum oriens was performed in the same region of interest (ROI). VGLUT1 and PSD-95 images were thresholded by a constant grayscale value equal to the average of the automatically calculated threshold level of all analyzed images. We measured the total intensity of all VGLUT1 or PSD-95 puncta in each ROI (total intensity (AU) per µm$^2$) and normalized the values to the mean value of the WT control.

### Dorsal root ganglion (DRG) and spinal cord section preparation

In all, 19-week-old TrkC KI mice and WT littermate mice were anesthetized with ketamine/xylazine mixture, followed by transcardial perfusion with 15 mL of 0.9% saline solution for blood removal and 15 ml of 4% PFA for fixation. Dissected lumbar vertebral columns (bones, spinal, and DRG tissues) were then post-fixed for 2 h at 4 °C with shaking. These were then transferred to PBS to dissect neuronal tissues with DRGs still attached to the spinal cord. The DRGs were then individually dissected and washed with PBS overnight at 4 °C with shaking. The spinal cord was placed back in 4% PFA and post-fixed overnight and washed in 1 M PBS the following day. Neural tissues were then incubated in 30% sucrose solution at 4 °C with shaking for 2–3 days for cryoprotection, embedded in O.C.T. compound, and finally stored at −80 °C until sectioning. Non-consecutive sections were collected for DRG samples (16-µm thick) and spinal cord samples (25-µm thick) using

a Cryostat (Thermo Scientific, CryoStar NX70), which were then left to dry at room temperature 1–2 h and then stored at −80 °C until use. For immunohistochemistry, slides were brought to room temperature and washed three times in 1 M PBS at room temperature, blocked for 30 min using blocking solution (1× PBS containing 5% heat-inactivated normal horse serum and 0.1% Triton X-100) and washed again three times in 1 M PBS. Samples were incubated overnight at 4 °C with a combination of the following primary antibodies/stain: anti-NeuN-Alexa 647 conjugated (1:1000, monoclonal rabbit, ab190565, Abcam), anti-NF200 (1:1000, polyclonal rabbit, N4142, MiliporeSigma), anti-TrkB (1:500, polyclonal goat, AF1494, R&D Systems), anti-TrkC (1:500, polyclonal goat, AF1404, R&D Systems), anti-CGRP (1:1000, polyclonal goat, ab36001, Abcam), anti-VGLUT1 (1:1000, polyclonal guinea pig, AB5905, Sigma-Aldrich) and IsolectinB4 stain-Alexa 647 conjugated (1:1000, I32450, Life Technologies). Following primary antibody incubation, slides were washed with PBS three times at room temperature. Samples previously incubated with unconjugated primary antibodies were then incubated for 2 h at room temperature with one of the following secondary antibodies: Alexa488-conjugated donkey anti-rabbit IgG (1:500, 711-545-152, Jackson ImmunoResearch) and Alexa488-conjugated donkey anti-guinea pig IgG (1:500, 706-545-148, Jackson ImmunoResearch), Alexa488-conjugated donkey anti-goat IgG (1:500, 706-545-147, Jackson ImmunoResearch). Slides were further washed with 1 M PBS three times and finally sealed with MOWIOL mounting medium.

### Synaptosome preparation

For the purification of crude synaptosome and synaptosome fractions from 4-week-old WT and TrkC KI littermate mice of mixed sex, we used a modification of a previously described method (Tanabe et al, 2017). Briefly, pooled hippocampi from each genotype were homogenized in cold buffer A (5 mM HEPES, pH 7.4, 1 mM MgCl$_2$, 0.5 mM CaCl$_2$, 1 mM DTT, 0.32 M sucrose, supplemented with protease inhibitors (Roche, 5892953001)), followed by centrifugation at 1400×$g$ for 10 min at 4 °C. The pellet was resuspended in buffer A, homogenized and centrifuged at 900×$g$ for 10 min at 4 °C, and the supernatant was combined with the initial supernatant (total lysate fraction). The combined supernatants were centrifuged at 12,000×$g$ for 10 min at 4 °C followed by resuspension of the pellets with cold buffer B (6 mM Tris, pH 8.1, 0.32 M sucrose, 1 mM EDTA, 1 mM EGTA, 1 mM DTT, supplemented with protease inhibitors) (crude synaptosome). The suspension was layered over a sucrose gradient (from bottom to top: 1.2 M, 1.0 M, 0.85 M sucrose, 6 mM Tris, pH 8.1) and centrifuged at 82,500×$g$ for 2 h at 4 °C. The synaptosome fraction was collected from the interface of the 1.0 and 1.2 M layers. Protein concentrations were then measured using DC protein assays (Bio-Rad).

### Co-immunoprecipitation

Crude synaptosomes from 4-week-old WT and homozygous KI littermate mouse brains were lysed in cold lysis buffer directly. The lysate was centrifuged for 10 min at 13,000×$g$ at 4 °C. The supernatant was pre-cleared and then incubated with Protein G Sepharose beads coated with anti-PTPσ antibody (monoclonal mouse, MM-0020-P, clone 17G7.2, MediMabs) overnight at 4 °C. After washing with lysis buffer three times, the samples were

denatured in SDS-loading buffer and run on 10% polyacrylamide gels for immunoblotting for TrkC and PTPσ.

## Total hippocampal lysate preparation

The hippocampi from 4-week-old WT and TrkC KI littermate were homogenized in cold 1× RIPA buffer (MilliporeSigma, 20-188) supplemented with protease inhibitors (Roche, 5892953001) and phosphatase inhibitors (Roche, 04906837001) and incubated for 40 min at 4 °C with rotation. The lysed samples were then centrifuged at 15,000× $g$ for 10 min at 4 °C. The supernatants were harvested, and the protein concentrations were then measured using DC protein assays (Bio-Rad).

## Immunoblotting

Samples were run on 10% polyacrylamide gels that were then transferred to PVDF membranes (Bio-Rad). The membranes were blocked in 5% skim milk and 0.1% Triton X-100 in PBS and incubated overnight with one of the following primary antibodies: anti-VGLUT1 (1:5000, polyclonal rabbit, 135302, Synaptic Systems), anti-PSD-95 (1:2000, monoclonal mouse, MA1-045, clone 6G6-1C9, Thermo Fisher Scientific), anti-TrkC (1:2000, monoclonal rabbit, 3376, clone C44H5, Cell Signaling Technology), anti-PTPσ (1:2000, monoclonal mouse, MM-0020-P, clone 17G7.2, MediMabs), anti-Neurotrophin-3 (1:500, polyclonal rabbit, ab53685, Abcam), anti-phospho-p44/42 MAPK (Erk1/2) (Thr202/Tyr204) (1:1000, polyclonal rabbit, 9101, Cell Signaling Technology), anti-p44/42 MAPK (Erk1/2) (1:1000, polyclonal rabbit, 9102, Cell Signaling Technology), anti-phospho-AKT (Ser473) (1:1000, monoclonal rabbit, 4060, Cell Signaling Technology), anti-AKT (1:1000, polyclonal rabbit, 9272, Cell Signaling Technology), anti-phospho-PLCγ1 (Ser1248) (1:1000, monoclonal rabbit, 8713S, Cell Signaling Technology), anti-PLCγ1 (1:1000, monoclonal rabbit, 5690T, Cell Signaling Technology), and anti-β-actin (1:2000, polyclonal rabbit, ab8227, Abcam) or anti-β-actin (1:1000, monoclonal mouse, ab8226, Abcam). Membranes were incubated with a corresponding horseradish peroxidase (HRP)-conjugated secondary antibody (1:5000, polyclonal donkey, 715-035-151 for donkey anti-mouse IgG (H + L) and 711-035-152 for donkey anti-rabbit IgG (H + L), Jackson ImmunoResearch). Signals were developed using Clarity™ Western ECL (Bio-Rad) and visualized using a ChemiDoc™ XRS+ Imager (Bio-Rad). Band intensities were corrected for background, measured using ImageJ, then normalized to β-actin intensity.

## Neurotrophin-3 (NT-3) binding experiments

To make the mammalian expression vector for mouse NT-3 tagged with a myc epitope controlled by the pCAG promoter (pCAG-NT-3-myc), the coding region of mouse NT-3 (NM_008742.2) tagged with a C-terminal myc epitope (SinoBiological, Cat: MG50223-CM) was subcloned into pCAG-EGFP (Addgene, Plasmid: #11150) between KnpI and NotI, replacing the EGFP coding region. The expression plasmids for extracellular HA-tagged CD4, TrkC WT, TrkC N366AT369A (NT-3-binding dead mutant) were described previously (Ammendrup-Johnsen et al, 2015). The expression plasmid for HA-TrkC D240A;D242A (PTPσ-binding dead mutant) was generated by inverse PCR and DpnI digestion.

To assess the binding of NT-3-myc to HA-TrkC constructs, COS-7 cells (RRID: CVCL_0224, ~30,000 cells per well) were co-transfected with 0.5 μg of pCAG-NT-3-myc and 0.5 μg of a plasmid expressing either HA-TrkC WT, HA-TrkC D270A;D272A, HA-TrkC

N366A;T369A, or HA-CD4 using TransIT-LT1 (Mirus Bio. LLC, MIR2305). Twenty-four hours after transfection, the cells were fixed in prewarmed 4% (v/v) formaldehyde and 4% (w/v) sucrose in PBS for 12 min. To label surface-bound NT-3-myc and surface-expressed HA-TrkC or HA-CD4, the fixed cells were blocked with a blocking solution (5% (v/v) normal donkey serum and 3% (w/v) bovine serum albumin in PBS) for 1 h at room temperature without cell permeabilization and then incubated with both anti-myc antibody (1:1000, polyclonal rabbit, 2272S; Cell Signaling Technology) and anti-HA antibody (1:1000, monoclonal mouse, 12CA5, Roche) overnight at 4 °C. Cells were then incubated with a mixture composed of Alexa594-conjugated donkey anti-rabbit IgG (1:500, 711-585-152, Jackson ImmunoResearch), Alexa488-conjugated donkey anti-mouse IgG (1:500, 715-545-151, Jackson ImmunoResearch). To label total NT-3-myc and total HA-TrkC or HA-CD4, the fixed cells were permeabilized in 0.2% (v/v) Triton X-100 in PBS and then immunolabeled for myc and HA. Images were acquired on a Leica DM6 fluorescence microscope with 20 × 0.7 numerical aperture objectives and a Hamamatsu C11440 ORCA-Flash 4.0 camera using LasX software (Leica).

Images were acquired as 16-bit grayscale while blind to the experimental condition. For quantification, sets of cells were stained simultaneously and imaged with identical settings. To quantify binding levels of NT-3-myc and cell surface expression levels of HA-tagged constructs, we measured the average intensity of each channel within the delineated COS-7 cell area subtracted by the average intensity of the off-cell background. For in situ binding assays, the average intensity of bound NT-3-myc was normalized using the average surface intensity of the HA-tagged protein signal. COS-7 cells expressing similar levels of HA-tagged proteins were selected to quantify bound NT-3-myc. Analyses were performed using Volocity 6.0, Excel for Microsoft 365 (Microsoft), and GraphPad Prism 10 (GraphPad Software).

## Hippocampal neuron culture, immunocytochemistry, and image analysis

Hippocampal neuron cultures were prepared from E18 mouse mixed-sex embryos as previously described (Kaech and Banker, 2006). In short, hippocampi from E18 mouse embryo littermates were extracted and stored individually in tubes containing Hibernate E (BrainBits, HE500) for a few hours. Meanwhile, genotyping of the mice was carried out as described in a previous section to identify WT and TrkC homozygous KI littermates. Hippocampi from littermates of the same genotype were pooled, incubated in 0.25% trypsin-EDTA and dissociated, then the cells were plated onto polylysine-coated coverslips. The next day, the coverslips were inverted onto 1-week-old rat glial cultures and incubated for 2 weeks.

Following the culture, the coverslips were removed and fixed for 12 min with parafix solution (4% paraformaldehyde and 4% sucrose in PBS, pH 7.4) followed by permeabilization with PBST (PBS + 0.2% Triton X-100). The coverslips were incubated overnight at 4 °C with a combination of the following primary antibodies: anti-VGLUT1 (1:2000, polyclonal guinea pig, AB5905, Sigma-Aldrich), anti-PSD-95 (1:1000, monoclonal mouse, MA1-045, clone 6G6-1C9, Thermo Fisher Scientific) and anti-MAP2 (1:3000, chicken polyclonal IgY, ab5392, Abcam). Following primary antibody incubation, coverslips were washed with PBS, then incubated for 1 h at room temperature with a mixture composed of Alexa594-conjugated donkey anti-guinea pig IgG (1:500, 706-585-148,

Jackson ImmunoResearch), Alexa488-conjugated donkey anti-mouse IgG (1:500, 711-545-151, Jackson ImmunoResearch) and AMCA-conjugated donkey anti-chicken IgY (1:200, 703–155–155, Jackson ImmunoResearch). Coverslips were finally washed with PBS and mounted onto slides with elvanol.

All imaging and image analysis was done while blind to the experimental condition. All the data for imaging were collected in random order. Analysis was performed using Metamorph 7.8. For immunocytochemistry analysis, neurons were chosen randomly based on similar cell density and morphology. After images were thresholded, synaptic protein puncta were delineated by the perimeter of the designated neuron. Three regions of dendrites per neuron were randomly selected, and the puncta size and number per dendrite length of synaptic protein puncta were measured. VGLUT1-positive PSD-95 clusters indicate the number of clusters with pixel overlap between the separately thresholded VGLUT1 and PSD-95 channels.

### Electron microscopy

Six littermates (3 from TrkC WT mice and 3 from TrkC KI mice, 4 weeks of age) were anesthetized and perfused transcardially with 2% paraformaldehyde and 2.5% glutaraldehyde in 0.1 M sodium cacodylate buffer (pH 7.4) for 15 min. Brains were removed and post-fixed in the same buffer at 4 °C overnight. Small block pieces (2 × 2 × 2.5 mm) of hippocampus including the CA1 stratum radiatum and stratum oriens were dissected and fixed once more in the same buffer at 4 °C overnight. Then, the pieces were subjected to post-fixation with 1% aqueous $OsO_4$ and 1.5% aqueous potassium ferrocyanide in the same buffer at 4 °C for 2 h. After dehydration with increasing concentrations of acetone, the samples were embedded in Epon812. To verify the location within the samples of the stratum radiatum or the stratum oriens, 500-nm-thick sections were prepared with an ultra-microtome and stained with toluidine blue. Ultrathin sections (70–80 nm thick) were cut using an ultra-microtome and stained with 4% uranyl acetate and Reynold's lead. The ultrathin sections on the TEM grids were imaged by an FEI Tecnai G2 Spirit TEM equipped with a Gatan Ultrascan 4000 CCD Camera Model 895 at an accelerating voltage of 120 kV. The original proprietary Digital Micrograph 16-bit images (DM3) were converted to unsigned 8-bit TIFF images. The final magnification of captured images was ×4800 to quantify synapse number and PSD length and ×9300 to quantify the number, distance and size of synaptic vesicles (SVs). Symmetric and asymmetric synapses and synaptic vesicles per bouton were visually identified and counted using Metamorph 7.8 (Molecular Devices). To measure the distance between SVs, the distance from an SV to its nearest neighbor was measured using the single line mode in the Metamorph. All images were acquired and analyzed while blind to genotype.

### Electrophysiology

TrkC KI mice and WT littermates (4–6 weeks old) underwent cervical dislocation after which whole brains were rapidly removed and placed in ice-cold cutting solution composed of (in mM): 210 sucrose, 11 D-glucose, 26.2 $NaHCO_3$, 2.5 KCl, 1 $NaH_2PO_4$, 0.5 $CaCl_2$, and 5 $MgSO_4$ with the pH equilibrated to 7.3–7.4 by 95% oxygen and 5% carbon dioxide and the osmolarity adjusted to 290–300 mOsm/L (unless stated, all chemicals and drugs were purchased from Sigma-Aldrich). Acute 400 μm slices containing dorsal hippocampus were obtained using a VT1200S vibratome

(Leica Biosystems) and then transferred to an incubation chamber (BSK12, Scientific Systems Design) containing a standard artificial cerebral spinal fluid (ACSF) comprised of (in mM): 124 NaCl, 10 D-glucose, 26 $NaHCO_3$, 2.5 KCl, 1.25 $NaH_2PO_4$, 2 $CaCl_2$ and 1 $MgSO_4$ at pH of 7.3–7.4 and osmolarity of 290–300 mOsm/L. ACSF was bubbled continuously with carbogen (95%$O_2$/5%$CO_2$). Slices recovered in the chamber at 32 °C for 40 min prior to experiments. Following additional recovery time at room temperature, slices were transferred to a submerged recording chamber (RC-26; Warner Instruments) mounted on an upright microscope (Nikon Eclipse FN1) and were continuously perfused with carbonated ACSF at a rate of 2–3 ml/min for the duration of the experiments.

Whole-cell patch-clamp recordings were made from pyramidal cells in the CA1 area of the hippocampus. The cells were identified using an upright microscope equipped with a ×40 water immersion objective and a CCD camera (DAGE IR-2000 CAMERA). Whole-cell voltage-clamp recordings were acquired using recording pipettes made from borosilicate glass electrodes (1.5 mm O.D. and 0.86 mm I.D., Sutter Instrument) filled with intracellular solution containing (in mM): 120 Cs-Gluconate, 10 CsCl, 0.3 Na GTP, 4 Mg ATP, 0.5 EGTA, 10 phosphocreatine-Na,10 HEPES, and 5 QX-314-Cl while the pH was adjusted to 7.3–7.4 and osmolarity was 290–300 mOsm/L. The tip resistance of the micropipettes pulled via a microelectrode puller (P-97, Sutter Instrument) was 3–5 MΩ. The whole-cell recording data were collected using a multiclamp 700B amplifier and digitized at 10 kHz with a Digidata 1550 A/D convertor and pClamp 11 software (Molecular Devices). The electrode capacitance and series resistance were compensated for, and recordings were rejected from further analysis if series resistance changed more than 25%. The liquid junction potential was not corrected.

For miniature EPSC (mEPSC) recordings, CA1 pyramidal neurons were voltage-clamped at −70 mV. During mEPSC recording, tetrodotoxin (TTX; 1 μM) and bicuculline (10 μM) were added to the ACSF to block action potentials and $GABA_A$ receptor-mediated inhibitory synaptic currents, respectively. The frequency and amplitude of mEPSCs were analyzed using MiniAnalysis software (Synaptosoft).

Evoked excitatory postsynaptic currents (eEPSCs) were recorded from pyramidal cells in the CA1 region while Schaffer collaterals were stimulated using a bipolar nickel-chromium electrode placed in the stratum radiatum. AMPAR-mediated currents were measured at a holding potential of −70 mV. To isolate NMDAR-mediated currents, eEPSCs were recorded at a holding potential of +40 mV. The peak NMDAR-mediated EPSC amplitude was then measured 35 ms after the peak amplitude of the EPSC recorded at −70 mV. Silent synapses were identified using coefficient of variance (CV) analysis. CV-NMDAR and CV-AMPAR were determined using the following formula:

$$CV = \sqrt{SV(EPSC) - SV(noise)}/mean(EPSC)$$

where SV (EPSC) represents the sample variation of the peak current amplitude, SV (noise) is the sample variation of the root mean square noise and mean (EPSC) is the average peak amplitude of the EPSCs. Once these values were extracted, the proportion of silent synapses was calculated as CV-NMDAR/CV-AMPAR.

Extracellular field excitatory postsynaptic potentials (fEPSPs) were recorded using an interface chamber (Scientific Systems

Design) and a MultiClamp 700B amplifier. fEPSPs were generated by stimulating the Schaffer collateral pathway using two bipolar nickel-chromium electrodes placed in the stratum radiatum area of the CA1 and recorded with a glass microelectrode filled with ACSF (resistance 2–3 MΩ). Input/output (I/O) curves were generated by monitoring changes in fEPSP slopes and fiber volley amplitudes in response to increasing stimulation intensities (10–100 μA). Four sweeps were recorded at each intensity with a 33 s interval and averaged to produce an accurate representation of responses at each intensity. Input/output curves were plotted as both a function of raw changes in fEPSP slopes and fiber volley amplitudes, as well as a percentage change compared to the average response of each intensity in that slice, to account for variability between slices. The presynaptic function was assayed through the application of paired pulses while varying the interpulse interval and calculated as a ratio of the second pulse as a percentage of the first (paired-pulse ratio; PPR). Extracellular recordings were acquired using Clampex 11.2. Analysis was done through a combination of Clampfit 11.2 and WinLTP 3.00. Two-way repeated measures ANOVA with Šídák's post hoc tests were used for I/O and PPR assays with significance set at $P < 0.05$.

### Mass spectrometry for quantitative proteomics and phosphoproteomics analyses

Synaptosomal fractions for proteomics and brain tissues for phosphoproteomics were homogenized in 6 M guanidine-HCl containing 100 mM HEPES−NaOH (pH 7.5), 10 mM Tris (2-carboxyethyl) phosphine (TCEP), and 40 mM chloroacetamide. After heating and sonication, proteins were purified by methanol −chloroform precipitation and solubilized with 0.1% RapiGest SF in 50 mM triethylammonium bicarbonate. After sonication and heating at 95 °C for 10 min, the proteins were digested with trypsin/Lys-C mix (Promega) at 37 °C overnight. For proteomics analysis, 20 μg of digested peptides from each sample were labeled with 0.1 mg of TMTpro 16-plex reagents (Thermo Fisher Scientific). For phosphoproteomics analysis, 250 μg of digested peptides from each sample were labeled with TMT 10-plex reagents (Thermo Fisher Scientific). Labeled peptides were enriched using a High-Select Fe-NTA Phosphopeptide Enrichment Kit (Thermo Fisher Scientific). The labeled peptides were pooled and fractionated using offline high-pH reversed-phase chromatography on a Vanquish DUO UHPLC system (Thermo Fisher Scientific) as previously reported (Yamanaka et al, 2023). Chromatographic separation was performed on a C18 reversed-phase column (4.6 × 250 mm Xbridge BEH130 C18 column with 3.5 μm particles; Waters) with a multistep gradient of mobile phase A (10 mM ammonium formate at pH 9.0 in 2% acetonitrile [ACN]) and B (10 mM ammonium formate pH 9.0 in 80% ACN). Peptides were separated into 48 fractions, which were consolidated into 16 fractions. Each fraction was evaporated in a SpeedVac concentrator and dissolved in 0.1% trifluoroacetic acid (TFA) and 3% ACN.

Liquid chromatography-tandem mass spectrometry (LC-MS/MS) analysis was performed on an EASY-nLC 1200 UHPLC connected to a Q Exactive Plus mass spectrometer through a nanoelectrospray ion source (Thermo Fisher Scientific). The peptides were separated on a C18 reversed-phase column (75 μm × 150 mm; Nikkyo Technos) with a linear gradient of 4–20% for 0–115 min and 20–32% for 115–160 min, followed by an increase to 80% ACN for 10 min and finally a hold at 80% ACN for 10 min.

The mass spectrometer was operated in data-dependent acquisition mode with a top 10 MS/MS method. MS1 spectra were measured with a resolution of 70,000, an AGC target of 3e6, and a mass range from 375 to 1400 *m/z*. MS/MS spectra were acquired at a resolution of 35,000, an AGC target of 1e5, an isolation window of 0.7 *m/z*, a maximum injection time of 150 ms, and a normalized collision energy of 33 (for proteomics analysis) or 34 (for phosphoproteomics analysis). Dynamic exclusion was set to 20 s.

Raw MS data were searched against the SwissProt database restricted to *Mus musculus* using Proteome Discoverer version 2.4 with the Sequest HT search engine. Analysis of MS spectra was performed using the following parameters: (a) trypsin as an enzyme with up to two missed cleavages; (b) precursor mass tolerance of 10 ppm; (c) fragment mass tolerance of 0.02 Da; (d) TMT of lysine and peptide N-terminus and carbamidomethylation of cysteine as fixed modifications; and (e) phosphorylation of serine, threonine and tyrosine (for phosphoproteomic analysis) and oxidation of methionine as variable modifications. Peptides were filtered at a false discovery rate (FDR) of 1% using the Percolator node. For the identification of phosphorylation sites, the threshold of localization probability calculated by the IMP-ptmRS node was set to 75%. TMT quantification was performed using the Reporter Ions Quantifier node. Normalization was performed such that the total sum of the abundance values for each TMT channel over all peptides was the same.

To isolate proteins differentially expressed and proteins differentially phosphorylated in TrkC KI mice and WT mice, using Perseus software version 2.0.3.1 (Tyanova et al, 2016), volcano plot analysis was performed applying cutoff criteria of an absolute fold change ($|FC|$) greater than 1.5-fold and a $P$ value less than 0.05 in two-tailed Student's $t$ tests. Additional statistical analysis for the confidence of the isolated data was performed using confidence criteria with permutation-based two-tailed Student's $t$ tests with 250 randomization, an FDR of 0.05 for the cortex or 0.1 for the hippocampus and s0 = 0.1.

For enrichment analyses, Gene Ontology analysis and Kyoto Encyclopedia of Genes and Genomes (KEGG) and were performed using gProfiler. UniProt IDs were submitted to the online platforms with the specification of *Mus Musculus* (mouse) datasets. SynGO analysis was performed using the open web software (https://syngoportal.org). Functional annotations and pathways with an FDR-corrected $P$ value ($q$ value) less than 0.05 were considered significant.

### Phos-tag electrophoresis with CIP treatment and blotting

Hippocampal tissues were lysed in a mixture of homogenization buffer (pH 7.4, 5 mM HEPES, 1 mM MgCl₂, 0.5 mM CaCl₂, 1 mM DTT, 0.32 M sucrose, protease inhibitors (Roche, 5892953001) and phosphatase inhibitors (Roche, 04906837001)). Calf-intestinal alkaline phosphatase (CIP, Promega, M2825) was used to verify the phosphorylation status of ITPKA; 15 units were added to the tissue lysate (1 μg/unit) at 37 °C for 1 hr. 15 μg/well tissue lysates from WT and TrkC KI samples were then loaded and subjected to electrophoresis on SuperSep™ Phos-Tag™ pre-cast gels (198-17981, FUJIFILM) consisting of a stacking gel (4.5% (w/v) acrylamide, 350 mM Bis-Tris, pH 6.8, 0.1% (v/v) N,N,N',N'-tetramethyl ethylenediamine (TEMED), and 0.05% (w/v) ammonium persulfate (APS)) and a separating gel (7.5% (w/v) acrylamide, 350 mM Bis-Tris, pH 6.8, 25 μM Phos-tag acrylamide, 100 μM ZnCl₂, 0.1% (v/v)

TEMED, and 0.05% (w/v) APS). Electrophoresis was performed at a constant current of ≤ 20 mA/gel with running buffer (100 mM Tris, 100 mM MOPS, and 0.1% (v/v) SDS) to which 5 mM sodium bisulfite was added immediately before electrophoresis. For immunoblot analysis, gels were pretreated by washing in methanol-free wash buffer (25 mM Tris, 192 mM glycine, 0.1% (v/v) SDS, 10 mM EDTA) six times for 10 min each to remove bivalent cations, followed by one wash in transfer buffer (25 mM Tris-Cl pH 7.4, 192 mM glycine, 10 mM EDTA) for 10 min. Then, the proteins were electroblotted onto polyvinylidene difluoride (PVDF) membrane in transfer buffer containing 5% (v/v) methanol and 1 mM EDTA at 25 V overnight. For efficient transfer of Phos-Tag™ gel separated proteins, we used a wet-tank transfer approach. Following electrophoretic transfer, membranes were allowed to dry completely. Subsequently, they were reactivated in methanol and immediately stained with 3% (w/v) Ponceau S in 5% (v/v) aqueous solution of acetic acid to validate transfer efficiency. Membranes were then thoroughly destained with MilliQ water and TBST (10 mM Tris-HCl (pH 7.5), 100 mM NaCl, and 0.10% (v/v) Tween-20). Non-specific antibody binding was blocked by incubating membranes in 6% (w/v) BSA (A9647-1006, Sigma-Aldrich) in TBST for at least 1 hr. Anti-ITPKA (polyclonal rabbit, PA5-75226, Thermo Fisher Scientific) was diluted at 1:500 in 6% w/v BSA in TBST, and incubation of the membranes was carried out overnight at 4 °C. Following extensive washing in TBST (3 × 10 min), membranes were incubated with HRP-conjugated secondary antibodies diluted at 1:5000 in 5% w/v skim milk in TBST for 1 hr. After washing in TBST (3 × 10 min) membranes were incubated with ECL reagent according to manufacturer's instructions. Immunoreactive bands were documented with a Bio-Rad ChemiDoc™ MP System. For quantification of β-actin as a loading control, the membrane was incubated with stripping buffer (pH 2.7, 25 mM Glycine-HCl, 1% (v/v) SDS) for 10 min. Non-specific antibody binding was blocked by incubating membranes in 5% w/v skim milk in PBST for 1 h. Anti-β-actin (monoclonal mouse, A5441, Sigma-Aldrich) was diluted at 1:5000 in 5% w/v skim milk in PBST and incubation of the membranes was carried out for 1 hr. Following washing in PBST (3 × 10 min), membranes were incubated with HRP-conjugated secondary antibodies diluted at 1:5000 in 5% w/v skim milk in PBST for 1 h. After washing in PBST (3 × 10 min) membranes were incubated with ECL reagent according to the manufacturer's instructions. Immunoreactive bands were documented with a Bio-Rad ChemiDoc™ MP System.

### Behavioral tests

Starting 14 days before the first day of behavioral tests, mice in their home cages were transferred from the housing room daily and placed in the experimental room for 1 h to habituate them to the novel environment and experimenter handling. Each time, 5 min of experimenter handling and habituation was performed. We used an extended habituation period, rather than the standard 3–5 days, because our preliminary tests showed that some mice exhibited enhanced escape behavior even after 1 week of habituation, and then we found that all of these mice were TrkC KI mice.

All behavioral experiments were carried out on age- and sex-matched WT and KI mice (2–4 months old) during similar daytime periods. The mice used for the behavioral tests were as follows: 3 cohorts for a total of 12 WT male, 12 TrkC KI male, 13 WT female and 14 TrkC KI female littermate mice for open field, Y-maze and

elevated plus maze; 2 cohorts for a total of 8 WT male, 8 TrkC KI male, 9 WT female and 10 TrkC KI female littermate mice for all the remaining tests detailed below. Before every experiment, mice were transferred and habituated to the test room for at least 30 min, with the room lighting already set as described in each test. All tests were recorded by a color camera with a GigE interface (Basler) and analyzed using EthoVision XT10 (Noldus), except for the marble burying test, which was quantified manually.

### Open-field test

Under overhead lighting at 150 lux, mice were placed into the center of an open square field (LxWxH: $50 \times 50 \times 38$ cm) at the start of the trial and exploration was recorded for 10 min. To assess locomotor activity, we measured total traveled distance and mean velocity. To assess anxiety level, time spent in the center area (40% of the total surface) was measured.

### Y-maze spontaneous alternation test

A Y-shaped maze with three opaque arms located 120 degrees apart (LxWxH: $35 \times 5 \times 20$ cm) under overhead lighting at 150 lux was used to evaluate mouse exploratory behavior. Mice were placed inside one arm facing away from the center and allowed to move through the apparatus for 8 min. Scoring consisted of recording each arm entry, defined as all four paws entering the arm.

### Three-chamber sociability and social novelty test

All procedures were performed with 50 lux lighting. The apparatus is composed of a three-compartment gray plexiglass arena ($L \times W \times H$: $60 \times 40.5 \times 22$ cm) containing two identical cages, one in each of the side compartments. For habituation, the test mouse was placed into the center chamber, and was allowed to explore the entire arena freely for 10 min to acclimatize to the arena and the empty cages. After this, the mouse was gently guided and restricted to the center chamber using the built-in doors for a 5 min rest period. To assess sociability, an age and sex-matched unfamiliar wild-type B6J mouse (stranger 1) was held within one of the cages, and the other cage contained a novel object. The test mouse was allowed to freely explore the entire arena for 10 min. After this session, the test mouse was once again guided and restricted into the center chamber for 5 min. Next, to assess social novelty, the novel object was replaced with a new unfamiliar wild-type B6J mouse (stranger 2). The test mouse was again allowed to freely explore all three chambers for 10 min. Prior to testing, stranger 1 and stranger 2 mice were habituated in the cages for 10, 15, and 20 min over 3 days. Furthermore, they originated from different breeding cages and had never interacted with each other or the test mouse prior to the tests or during cage habituation period. The placement of mice and object cages were counterbalanced (left or right chamber) between different experimental mice. In both sessions, the trial was recorded and then scored with EthoVision XT10 for the time the subject mouse spent in close proximity and/ or sniffed the cage containing the stranger 1 mouse and the neutral object in the sociability session and the stranger 1 (familiar) and stranger 2 mice (novel) in the social novelty session.

### Contextual and cued fear conditioning test

Fear conditioning was performed as previously described (Xu et al, 2012) with slight modifications. Briefly, on the training day, mice were placed in the fear conditioning chamber located in the center

of a sound-attenuating cubicle. A tissue paper soaked with 10% ethanol was placed within the conditioning chamber to provide a background odor and white noise at 65 dB provided a background noise. After a 2-min exploration period, three tone-footshock pairings were delivered at 1-min intervals. The 95 dB 2 kHz tone lasted for 30 s, with the 0.75 mA footshocks being activated during the last 2 s and terminating with the tone. After the final tone-footshock pairing, the mice remained in the training chamber for another 30 s before being returned to their home cages. The context test occurred 24 h later with mice placed back into the original conditioning chamber for 5 min. For the altered-context and tone tests, the test room surroundings were changed, and the conditioning chamber was modified by changing its metal grid floor to a plastic sheet and the transparent plexiglass side walls to plastic walls with black stripes, the background ethanol odor was changed to vanilla and the background noise was turned off. Mice were placed in the altered chamber 24 h after the context test for a 5 min period, after which a tone (95 dB, 2 kHz) was delivered for 1 min. During the whole 3-day process, the measured output was the freezing time of the mice in the different contexts, defined as the absence of movement lasting more than 1 s. To record and analyze mouse behaviors, the Ugo Basile 46000 Fear Conditioning System and EthoVision XT17 software were used.

### Light and dark transition test

The test was conducted in an arena (L × W × H: $44 \times 27 \times 27$ cm) consisting of two compartments of unequal size. The lit compartment constituted the larger compartment (L × W: $27 \times 27$ cm) and was constructed of white plexiglass with no ceiling. The dark compartment constituted the smaller compartment (L × W: $18 \times 27$ cm) and was made of black plexiglass, covered with a black plexiglass ceiling. A small opening ($7.5 \times 7.5$ cm) located midline between the two compartments allows access between the two compartments. The lit compartment was illuminated at over 300 lux using room lighting as well as overhead LED white lamps. The mouse was placed in the dark compartment and allowed to explore the arena for a 10-min test period. Latency to first enter the light compartment and total duration of time spent in the light compartment were measured.

### Marble burying test

Testing was conducted under dimmed light (15 lux) in a larger novel cage (L × W × H: $47.5 \times 25.5 \times 20$ cm), filled with corncob bedding to a depth of ~5 cm and covered with a transparent plexiglass lid. Twenty marbles were evenly placed in the test cage in a $4 \times 5$ grid pattern. Before testing, mice were never exposed to the marbles. Mice were exposed to the marbles for 30 min, before being returned to their home cages without disturbing the marbles. Three different experimenters blind to genotype independently counted the number of marbles buried (at least 75% submerged with bedding).

### Sucrose splash test

Mice in home cages were acclimated to the behavioral room under dimmed lightning (15 lux) for 1 hr. Littermates from the same cage were transferred to a new temporary cage and left to habituate for a further 30-min period. To perform splash tests, one of the mice from the temporary cage was transferred back to its home cage and sprayed once with the 10% sucrose solution on the dorsal coat of the mouse. Then, the mouse was left undisturbed in the cage for 10 min while recording. After recording, the mouse was transferred

to a separate temporary cage, and the process was repeated one mouse at a time until tests were done for all mice. Grooming behaviors were automatically measured by the Mouse Behavior Recognition Module of EthoVision XT.

### Elevated plus maze test

The elevated plus maze consists of a maze at a height of 50 cm above the floor, formed in the shape of a plus (+) sign, with two of the arms having walls (closed arms), whereas the remaining two arms are without walls (open arms). Under overhead lighting at 150 lux, the mouse was placed in the middle of the elevated plus maze apparatus facing one of the closed arms and allowed to freely explore the maze for 5 min. The number of entries into and total time spent in the closed and open arms was recorded and measured.

### Statistical testing

Analyses were performed by using Metamorph 7.8 (Molecular Device), Microsoft Excel, and GraphPad Prism 10 (GraphPad Software). No statistical methods were used to predetermine the sample size. The data distribution was assumed to be normal. Statistical comparisons were made with two-tailed Student's unpaired $t$ tests, one sample $t$ tests, one-way ANOVA with post hoc Šídák's multiple comparison tests, two-way repeated measures ANOVAs with Šídák's post hoc tests, Kolmogorov–Smirnov tests or Kruskal–Wallis with Dunn's post hoc tests, as indicated in the figure legends. All data are reported as the mean ± standard error of the mean (SEM). No statistical significance (ns) was defined as $P > 0.05$, while statistical significance was defined as $*P < 0.05$, $**P < 0.01$, and $***P < 0.001$.

## Data availability

The MS proteomics data in this study have been deposited to the ProteomeXchange Consortium via the jPOST partner repository with the dataset identifiers PXD052653 (Fig. 5A,B), PXD052654 (Fig. 5C), and PXD052655 (Fig. 5D). The data that support the findings of this study are available within the article and its supplementary materials, or from the lead contact, H Takahashi (Hideto.Takahashi@ircm.qc.ca) upon request.

The source data of this paper are collected in the following database record: biostudies:S-SCDT-10_1038-S44318-024-00252-9.

## Peer review information

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

## Acknowledgements

The authors thank Johanne Ouellette and Dr. S Kelly Sears at the Facility for Electron Microscopy Research at McGill University for technical support for the EM study and Dr. Kohei Nishino at the Technical Support Department of Tokushima University for support for proteomic analysis. This work was supported by the Natural Sciences and Engineering Research Council of Canada (NSERC) Discovery grant (RGPIN-2017-04753), the Canadian Institutes of Health Research grants (MOP-133517, PTJ-191947), National Institutes of Health (NIH) (NIMH R01MH077303) and Fonds de la Recherche du Québec Research Scholars (FRQS Junior 2 (29106) and senior (251655)) grants to HT, the JSPS KAKENHI (JP22H02970) to MT, and the Research Cluster program of Tokushima University to MT, MI, and HKo, the Canada Research Chair (CRC) award and NSERC grant (RGPIN-2021-03612) to SC, the FRQS doctoral scholarship (303256), and the IRCM doctoral and Emmanuel-Triassi scholarships to HKh, the JSPS KAKENHI (JP21K15314) and HIRAKU-Global Program by MEXT's "Strategic Professional Development Program for Young Researchers" funding to MI, the Ontario Graduate Scholarship (OGS) to KP, the Iizuka Takeshi Scholarship Foundation to YN, the FRQS doctoral award (252652) to AKL, the IRCM doctoral scholarship to NC, and the Alzheimer Society Research Program doctoral award to NY.

## Author contributions

**Husam Khaled**: Conceptualization; Resources; Formal analysis; Validation; Investigation; Visualization; Methodology; Writing—original draft; Writing—review and editing. **Zahra Ghasemi**: Formal analysis; Investigation; Visualization; Methodology; Writing—review and editing. **Mai Inagaki**: Resources; Formal analysis; Validation; Investigation; Visualization; Writing—review and editing. **Kyle Patel**: Formal analysis; Investigation; Visualization; Writing—review and editing. **Yusuke Naito**: Resources; Investigation. **Benjamin Feller**: Formal analysis; Investigation; Writing—review and editing. **Nayoung Yi**: Investigation; Methodology; Writing—review and editing. **Farin B Bourojeni**: Formal analysis; Investigation; Writing—review and editing. **Alfred Kihoon Lee**: Investigation. **Nicolas Chofflet**: Investigation. **Artur Kania**: Writing—review and editing. **Hidetaka Kosako**: Formal analysis; Writing—review and editing. **Masanori Tachikawa**: Supervision; Investigation; Visualization; Writing—review and editing. **Steven Connor**: Formal analysis; Supervision; Visualization; Writing—review and editing. **Hideto Takahashi**: Conceptualization; Resources; Formal analysis; Supervision; Funding acquisition; Investigation; Visualization; Writing—original draft; Project administration; Writing—review and editing.

Source data underlying figure panels in this paper may have individual authorship assigned. Where available, figure panel/source data authorship is listed in the following database record: biostudies:S-SCDT-10_1038-S44318-024-00252-9.

## Disclosure and competing interests statement

The authors declare no competing interests.

