## [Peer Review File · The EMBO Journal]

The TrkC-PTP σ complex governs synapse maturation and angiogenic avoidance via synaptic protein phosphorylation

Husam Khaled, Zahra Ghasemi, Mai Inagaki, Kyle Patel, Yusuke Naito, Benjamin Feller, Nayoung Yi, Farin Bourojeni, Alfred Lee, Nicolas Chofflet, Artur Kania, Hidetaka Kosako, Masanori Tachikawa, Steven Connor, and Hideto Takahashi

Corresponding authors: Hideto Takahashi (Hideto.Takahashi@ircm.qc.ca) , Steven Connor (sac Connor@yorku.ca), Masanori Tachikawa (tachikaw@tokushima-u.ac.jp)

Review Timeline:

Submission Date:	1st Feb 24
Editorial Decision:	6th Mar 24
Revision Received:	8th Jul 24
Editorial Decision:	12th Aug 24
Revision Received:	26th Aug 24
Accepted:	30th Aug 24

Editor: Kelly Anderson

Transaction Report:

Dear Dr. Takahashi,

Thank you for submitting your manuscript for consideration by the EMBO Journal. It has now been seen by two referees whose comments are shown below.

Given the referees' positive recommendations, I would like to invite you to submit a revised version of the manuscript, addressing the comments of both reviewers. I should add that it is EMBO Journal policy to allow only a single round of revision, and acceptance of your manuscript will therefore depend on the completeness of your responses in this revised version. It would be good to discuss you plan to address the referee concerns and I am available to do so by zoom or email in the coming weeks.

Thank you for the opportunity to consider your work for publication. I look forward to your revision.

Yours sincerely,

Kelly M Anderson, PhD
Editor, The EMBO Journal
k.anderson@embojournal.org

We realize that it is difficult to revise to a specific deadline. In the interest of protecting the conceptual advance provided by the work, we recommend a revision within 3 months (4th Jun 2024). Please discuss the revision progress ahead of this time with the editor if you require more time to complete the revisions.

Referee #1:

In this study, Khaled and colleagues investigate the consequences of disrupting the interaction of the postsynaptic Ntrk3/TrkC receptor with the presynaptic PTPsigma receptor. To do so, the authors generated a knockin (KI) mouse model in which two key residues for the interaction with PTPsigma, identified in a previous study (Coles et al., Nat Commun 2014), have been substituted for alanines in the extracellular domain of TrkC. The authors perform an extensive characterization of the TrkC KI mouse at the immunohistochemical, electron microscopic (EM), electrophysiological, proteomic, and behavioral level. Main observations are an increase in presynaptic Vglut1 intensity, increased clustering of synaptic vesicles and elongation of postsynaptic density, a decreased vesicle release probability at presynaptic terminals, altered phosphorylation of primarily synaptic proteins, and behavioral defects suggesting impaired social novelty and increased avoidance of anxiogenic conditions such as the center of an open field. The findings are interesting, the quality of the data is convincing, and experiments are well described and presented, making this a study that is easy to follow. The study could be improved at the mechanistic level: phenotypic analyses are well done but no effort is made to experimentally connect observations at the EM and electrophysiological level for example, or to link synaptic phenotype to proteomics/behavioral observations. Strengthening these links with experimental data would improve the paper.

Major

1. I could not find evidence in the Coles et al., 2014 study that the D240A; D242A substitutions in TrkC do not affect neurotrophin NT3 binding to TrkC. I agree with the authors that these amino acid substitutions are in a receptor domain that is different from the NT3 binding domain, but if this indeed has not been shown before it will be important to demonstrate that NT3 binding to the TrkC D240A; D242A receptor is not affected. Related to this, TrkC knockout (KO) is known to affect sensory neuron number in dorsal root ganglia (DRG) and myelinated axon number in dorsal root (Ménard et al., Dev Biol 2018; Klein et al., Nature 1994). It would be good to assess whether these parameters, DRG numbers and myelinated axon numbers, are indeed unaltered in TrkC KI mice to confirm that NT3-dependent function is unaffected.
2. EM analysis shows increased clustering of synaptic vesicles and elongated postsynaptic density (Figure 3). Electrophysiological analysis shows an increase in basal synaptic transmission and decrease in presynaptic release probability (Figure 4). How these observations are linked does not become clear. Do the authors think that increased clustering of vesicles leads to reduced release probability? Is distance to the active zone altered in TrkC KI presynaptic terminals? The authors speculate in the Results that the electrophysiological phenotype may result from upregulation of postsynaptic function. This can be assessed by additional analysis of spontaneous synaptic transmission (mEPSC frequency and amplitude) and by investigating AMPA receptor and NMDA receptor mediated evoked EPSCs to provide insight in how EM and electrophysiological observations are correlated. This can be complemented with analysis of AMPAR, NMDAR levels at TrkC KI hippocampal synapses. It could also be insightful to compare functional parameters for stratum radiatum and stratum oriens, as postsynaptic measures (IHC and EM) differ for stratum radiatum vs. oriens.
3. The phosphoproteomics experiment (Figure 6) shows many synaptic hits with altered phosphorylation. The authors speculate in the Discussion (page 14) that phosphorylation of several of these molecules and their regulators is altered in TrkC KI mice, but it is not clear which ones they mean: are synapsin I, bassoon and piccolo phosphorylation altered in their dataset? Based on their results it should be possible to come up with a hypothesis that can be tested experimentally to demonstrate how altered phosphorylation levels of the synaptic proteins identified here alters pre- or postsynaptic properties. Validation experiments of the phosphoproteomics dataset would complement the observations but I understand that this will depend on availability of antibodies against phosphoepitopes for candidate proteins.

Minor

1. Fig. 1D: what does M stand for?
2. Fig. 2A-C: is Vglut1 and PSD95 puncta number and overlap affected in TrkC KI mice?
3. Fig. 2B: there appears to be a bimodal distribution in both the Vglut1 and PSD95 condition. Please explain.
4. Fig. 2B-C: data points are ROIs; please perform a nested analysis so data points are mice. Similar for Fig. 3B.
5. Fig. 2D: also show a representative image.
6. Fig. 2F: what is shown in graph, number of mice?
7. Fig. 5C-D: how are the thresholds chosen?
8. Fig. 5C-D and 6A-B: what was rationale for including cortex instead of focusing on hippocampus as was done in the previous figures?
9. The link between phosphoproteomics and behavior is not particularly strong. Can a statistical test be done for the SFARI

associated proteins to determine whether these are over-represented in the dataset?

10. As a suggestion, Figure 1 and 5 could be moved to supplemental figures; behavioral data could be condensed in one main and one supplemental figure.

11. The Discussion is well written and integrates the results in the existing literature. I missed discussion of previous studies assessing synaptic parameters in TrkC KO mice. Martínez et al., J Neurosci 1998 analyze synaptic density and presynaptic morphology in TrkC KO using EM; there may be additional studies. Similarities and differences with the TrkC KI will be interesting to discuss.

Referee #2:

In the present report, Khaled and colleagues generate and analyze a mouse knock-in model that disrupts the trans-synaptic interaction of TrkC with the post-synaptic receptor PTPsigma. In this elegant genetic model, the neurotrophic function of TrkC with the neurotrophin NT-3 is preserved, which allows to dissect the impact of the trans-synaptic adhesion on synapse organization, function and their behavioral consequences. With this model, the authors show that TrkC/PTPsigma interaction regulates the phospho-proteome of synapses, the excitatory presynaptic organization, the excitatory neurotransmission and alters social behavior. This is of major importance for several fields. The knock-in approach revealing multifunctional aspects of single proteins is of interest for most scientists interested in the structure/function analysis of proteins. The analysis of TrkC functions is of broad interest for cell biologists. The analysis of the trans-synaptic adhesion system is of broad interest for molecular neurobiologists. The behavioral consequences of the mutation in relation to psychopathology is of broad interest for neuroscientists and clinicians.

However, several aspects of the work may request further experiments or major corrections prior to publication. The overall impression is that many topics have been investigated, but not enough to reach firm and reliable conclusions.

Major points :

1/ VGLUT1 expression / distribution

The authors provide immunofluorescence data that strongly suggest a global increase of expression of VGLUT1 both in-vivo and in-vitro in the TrkC-KI. Based on an immunoblot on synaptosomes that shows no difference of expression of VGLUT1, the authors conclude that TrkC-KI leads to a relocation of VGLUT1 but not to a change of expression level. There is a need for additional experiments on this point to reach a conclusion that is fully supported by the data. Indeed, the authors immunostainings reveal an increase of signal that is difficult to reconcile with a relocation of the proteins. Furthermore, the images from cultured neurons do not suggest a change of distribution. Therefore, either the authors provide additional data showing direct evidence for a change of trafficking of VGLUT1 or they have to show a consistent set of data showing the impact of TrkC-KI on the expression level of VGLUT1. In general, a set of immunoblots of an array of synaptic proteins on homogenates of hippocampus and/or cortex may also be helpful.

For PSD-95 the dataset is more coherent because the authors show a more local change of expression in the stratum radiatum layer of the hippocampus that can be explained by the increased PSD size measured in electron microscopy only in that layer. This increase may escape a global immunoblot analysis for the whole hippocampus.

2/ Ultrastructural changes at excitatory hippocampal synapses of TrkC KI

The authors provide a thorough analysis of the TrkC KI compared to WT using transmission electron microscopy.

- The measurement of synapse density is correct in my opinion and comforts the idea that synapse density is unchanged.
 - The measurement of PSD size seems appropriate and displays a local increase in stratum radiatum. A further analysis of this phenotype may be of interest. Are other post-synaptic proteins and receptors regulated in this region? What is the functional impact?
 - The measurement of synaptic vesicle distribution and density is more problematic in my opinion, because it is now broadly accepted that chemical fixation methods alter SV distribution in the terminals (see for instance Siksou, J Neurosci, 2007; Siksou, EJN, 2013; Korogod, elife, 2015). Alternatively, high pressure freezing provide a much better tissue preservation and quality for this type of analysis. Data such as the number of docked/primed SVs is accessible for instance. Hence, the present data suggests that the SV cluster is more compact, but it is likely a joint consequence of relevant cellular alterations that change the dynamics of SV mobility induced by the KI together with irrelevant crosslinking induced by the chemical fixation. The authors must take this into account to revise their conclusion or perform high pressure freezing experiments to produce a complementary data set helping to draw firm conclusions.
 - In figure 3E, SV density measurements are displayed by number of SV/bouton, this might induce a bias as the measurements were done on single thin sections and not by serial sections or 3D tomography. Instead, the area of boutons where SVs are counted should be used for normalization. The results will be displayed as SV number per μm^2 of bouton area.
- Altogether, the EM analysis is of high potential interest with respect to the observation of VGLUT1 and PSD95 expression and phosphor-proteomic alterations, but the methodology should be revised to reach reliable conclusions.

3/ Proteomics analysis

The authors provide 2 proteomics datasets comparing TrkC KI and WT. A global quantification of proteins of the hippocampal synaptosomes and a phosphor-proteomic analysis of the same sample using labeling methods.

- The global assay shows no significant difference between genotypes. This is most likely due to the fact that synaptosome preparations are quite bulk compared to the changes that occur only at some excitatory synapses in this KI model. I'm not sure

that it is wise to conclude that no changes of protein amounts occur in this mouse model from this experiment on bulk synaptosomes

- The phosphor-proteome dataset shows some potentially interesting differences in the phosphorylation of proteins involved in synapse biology and may have a link with psychiatric behavioral dimensions. Yet, a validation of some of these potentially interesting targets with independent methods is needed to strengthen the finding. A connection between altered phosphorylation of some proteins and the phenotypes of SV dynamics could be investigated to clarify some opened questions of the current study.

- Some important information is missing to allow for a proper evaluation of both datasets. Like for instance how many proteins were identified, Criteria for inclusion in the differential quantification, coverage of the syngo database...

4/ Behavioral assessment

The authors provide very interesting phenotypes from the assays performed. Yet a thorough SHIRPA set of tests would be nice to make sure that no major phenotype escaped the attention of the investigators.

5/ Altogether, it is unclear at the end of the reading how the authors link the mutation they induced to the alteration of VGLUT1, PSD95, synapse ultrastructure, synapse phosphor-proteome and mouse behavior. While some interesting but incomplete evidence are provided, causal links are often missing.

Minor points:

1/ Grouping the KI validations in figure 1 may be better than having some aspects in other figures like Fig 5AB or Fig S4

2/ Positive control for NT3 signaling

While the KI mice seem to be unaffected for NT3 signal transduction, it seems that the article lacks an assay that measures this in a quantitative and dynamic way.

Authors' response to the reviewers' comments for the manuscript **EMBOJ-2024-116855** of Khaled et al., **“The TrkC-PTP σ complex governs synapse maturation and anxiogenic avoidance via synaptic phosphorylation”**, and description of changes made in the revised paper

First of all, we thank all the reviewers for their careful reviewing and helpful suggestions to improve the manuscript. We performed additional experiments to answer the reviewers' questions and to strengthen our conclusions. These new data have been integrated into our revised manuscript. The point-by-point answers are below with the reviewers' comments in *italic* font and the responses and descriptions of changes made in the manuscript in **bold font**.

Referee #1:

In this study, Khaled and colleagues investigate the consequences of disrupting the interaction of the postsynaptic Ntrk3/TrkC receptor with the presynaptic PTPsigma receptor. To do so, the authors generated a knockin (KI) mouse model in which two key residues for the interaction with PTPsigma, identified in a previous study (Coles et al., Nat Commun 2014), have been substituted for alanines in the extracellular domain of TrkC. The authors perform an extensive characterization of the TrkC KI mouse at the immunohistochemical, electron microscopic (EM), electrophysiological, proteomic, and behavioral level. Main observations are an increase in presynaptic Vglut1 intensity, increased clustering of synaptic vesicles and elongation of postsynaptic density, a decreased vesicle release probability at presynaptic terminals, altered phosphorylation of primarily synaptic proteins, and behavioral defects suggesting impaired social novelty and increased avoidance of anxiogenic conditions such as the center of an open field. The findings are interesting, the quality of the data is convincing, and experiments are well described and presented, making this a study that is easy to follow. The study could be improved at the mechanistic level: phenotypic analyses are well done but no effort is made to experimentally connect observations at the EM and electrophysiological level for example, or to link synaptic phenotype to proteomics/behavioral observations. Strengthening these links with experimental data would improve the paper.

(Response) We thank the reviewer for the overall positive evaluation of our study and the comment that **“The findings are interesting, the quality of the data is convincing, and experiments are well described and presented, making this a study that is easy to follow.”** We hope that the additional data in the revised manuscript significantly address the reviewer's concerns to strengthen the conclusion.

Major

1. I could not find evidence in the Coles et al., 2014 study that the D240A; D242A substitutions in TrkC do not affect neurotrophin NT3 binding to TrkC. I agree with the authors that these amino acid substitutions are in a receptor domain that is different from the NT3 binding domain, but if this indeed has not been

shown before it will be important to demonstrate that NT3 binding to the TrkC D240A; D242A receptor is not affected. Related to this, TrkC knockout (KO) is known to affect sensory neuron number in dorsal root ganglia (DRG) and myelinated axon number in dorsal root (Ménard et al., Dev Biol 2018; Klein et al., Nature 1994). It would be good to assess whether these parameters, DRG numbers and myelinated axon numbers, are indeed unaltered in TrkC KI mice to confirm that NT3-dependent function is unaffected.

(Response) We agree with the reviewer that it is important to show that the NT3 binding is unaffected by the TrkC point mutations. To address this, we carried out cell surface protein binding assays. We found that the TrkC D240A;D242A (PTP σ -binding dead) mutant binds to NT-3 at a level comparable to TrkC wild-type (WT), while the TrkC N366A;T369A (NT-3-binding dead) mutant fails to bind to NT-3, indicating that the D240A;D242A substitutions in TrkC do not affect neurotrophin NT3 binding to TrkC (Appendix Fig. S1). We further performed co-IP experiments using total brain lysates to assess endogenous TrkC-NT-3 binding, but, unfortunately, we couldn't detect NT-3 signals even in WT samples presumably due to NT-3 being a small soluble protein and the lack of good antibodies to detect endogenous NT-3 in co-IP samples (we attempted the experiment using several commercial antibodies with no success).

With regards to the reviewer's comment suggesting that we should assess several parameters in sensory neurons to confirm that NT-3-dependent functions are unaffected, we have carried out immunohistochemical analysis on lumbar DRG slices as well as spinal cord slices (Appendix Fig. S9). As shown in the figure, there was no significant difference in the number of either total sensory neurons or myelinated neurons between KI and WT samples. Similarly, none of the afferent markers in the spinal cord slices showed any difference between WT and KI. These data confirm that the TrkC KI mice indeed retain normal NT-3-dependent functions.

Furthermore, we performed WB experiments using hippocampal total lysates and found that phosphorylation levels of NT-3-signaling-related molecules such as MAPK1/3, AKT and PLC γ in TrkC KI mice are comparable to those in WT (Appendix Fig. S8C). These data further support that the D240A; D242A substitutions in TrkC do not affect NT-3-related signaling.

2. EM analysis shows increased clustering of synaptic vesicles and elongated postsynaptic density (Figure 3). Electrophysiological analysis shows an increase in basal synaptic transmission and decrease in presynaptic release probability (Figure 4). How these observations are linked does not become clear. Do the authors think that increased clustering of vesicles leads to reduced release probability? Is distance to the active zone altered in TrkC KI presynaptic terminals? The authors speculate in the Results that the electrophysiological phenotype may result from upregulation of postsynaptic function. This can be assessed by additional analysis of spontaneous synaptic transmission (mEPSC frequency and amplitude) and by investigating AMPA receptor and NMDA receptor mediated evoked EPSCs to provide insight in how EM and electrophysiological observations are correlated. This can be complemented with analysis of

AMPA, NMDAR levels at TrkC KI hippocampal synapses. It could also be insightful to compare functional parameters for stratum radiatum and stratum oriens, as postsynaptic measures (IHC and EM) differ for stratum radiatum vs. oriens.

(Response) We agree that in the previous manuscript, the linkage between EM phenotypes and fEPSP phenotypes remained unclear. To make it clearer and given the reviewer's suggestions, we performed two additional electrophysiology experiments using whole cell patch-clamp recording of hippocampal CA1 pyramidal cells: (1) we assessed AMPAR- and NMDAR-mediated EPSCs evoked by Schaffer collateral (SC) stimulation, and (2) we assessed AMPAR-mediated miniature EPSCs (mEPSCs).

In the evoked EPSC experiments (Fig. 4A-C), we first quantified the AMPAR/NMDAR ratio and found that this is lower in TrkC KI synapses than in WT synapses, suggesting that TrkC KI synapses may show impairment of synapse maturation. We further quantified the CV-NMDAR/CV-AMPA ratio, which allows us to measure silent synapses (NMDAR-positive but AMPAR-negative synapses) and found that this is also lower in TrkC KI synapses, suggesting that there is a larger number of silent synapses in TrkC KI mice than in WT mice. Given our EM data showing no difference in total excitatory synapse number in the st. rad of TrkC KI and WT mice, the evoked EPSC data also suggest that TrkC KI mice have a smaller number of active synapses (NMDAR- and AMPAR-positive synapses) than WT mice.

We next performed AMPAR-mediated mEPSC experiments to assess the function of active synapses by quantifying mEPSC frequency and amplitude (Fig. 4D-F). Very interestingly, we observed a significant increase in mEPSC frequency in TrkC KI synapses. On the other hand, mEPSP amplitude was comparable between TrkC KI and WT synapses. These data suggest that active synapses in TrkC KI mice may show increased glutamate releasing sites and/or increased release probability without changing postsynaptic AMPARs.

Our data about AMPAR-mediated fEPSPs already revealed that TrkC KI SC-CA1 synapses show a significant increase in fEPSP slopes in I/O curves and significant enhanced paired-pulse facilitation, which suggests impaired release probability (Fig. 4G-J). Integrating this with our new mEPSC results suggests that active synapses in TrkC KI mice may have more releasing sites, which is consistent with increased fEPSP slopes as well as increased mEPSC frequency, but impaired probability of glutamate release from the releasing sites, especially under action potential-dependent release conditions.

Given these new results, we revised the Results section describing the electrophysiology experiments to conclude that TrkC KI substitution increases silent synapses (Page 7, line 227-230) and causes aberrant active synapses with abnormally enhanced basal synaptic transmission (presumably due to increased glutamate releasing sites) but lower release probability than WT active synapses (Page 8, line 251-255). We further added a discussion that such aberrance of

active synapses in TrkC KI mice might be compensatory to delayed synapse development and increased silent synapses (Page 15-16, line 519-538).

3. The phosphoproteomics experiment (Figure 6) shows many synaptic hits with altered phosphorylation. The authors speculate in the Discussion (page 14) that phosphorylation of several of these molecules and their regulators is altered in TrkC KI mice, but it is not clear which ones they mean: are synapsin I, bassoon and piccolo phosphorylation altered in their dataset? Based on their results it should be possible to come up with a hypothesis that can be tested experimentally to demonstrate how altered phosphorylation levels of the synaptic proteins identified here alters pre- or postsynaptic properties. Validation experiments of the phosphoproteomics dataset would complement the observations but I understand that this will depend on availability of antibodies against phosphoepitopes for candidate proteins.

(Response) To make it clear which ones we mean, we revised the appropriate sentence as shown below.

Page 15, line 508-510: “Importantly, our phosphoproteomic analyses have shown that phosphorylation of several of these molecules and/or their regulators, including piccolo Caps-1 (Cadps), dynamin-1 and rabphilin 3A, is altered in TrkC KI mice. ”.

As the reviewer pointed out, it is challenging to validate phosphoproteomics datasets by standard Western blot (WB) experiments because of the lack of antibodies that detect phosphorylation of the identified sites. We used an alternative approach based on Phos-tag WB (PMID: 26676139). Phos-tag traps phosphorylated proteins and allows us to distinguish between the phosphorylated and non-phosphorylated forms of the protein of interest. Of the proteins isolated in our phosphoproteomics experiment, we selected ITPKA because it had one of the highest fold changes in hippocampal samples and all phosphopeptides of ITPKA in TrkC KI mice were hyperphosphorylated compared to WT. Phos-tag WB using an anti-ITPKA antibody confirmed the hyperphosphorylation of ITPKA in TrkC KI mice (Appendix Fig. S6).

In addition, as mentioned above, our WB experiments show no change in the phosphorylation NT-3 signaling molecules, consistent with our phosphoproteomic results (Appendix Fig.S8).

To demonstrate how altered phosphorylation levels of the synaptic proteins identified here alters pre- or postsynaptic properties, we attempted a cell-biological experiment to mimic the phosphorylation changes in dynamin-1 that we observed in our phosphoproteomic experiments (Fig. 5: S853;S857 hyperphosphorylation in both hippocampal and cortical samples as well as S774D;T776D hyperphosphorylation in the hippocampal samples). We therefore induced neuronal overexpression expression of WT or phosphomimetic dynamin-1 (S853D;S857D, S774D;T776D and S774D;T776D;S853D;S857D) and then assessed VGLUT1 puncta because TrkC KI cultured neurons show an increase in VGLUT1 signals. However, we found that overexpression of any dynamin1 construct (including WT) resulted in significant reduction of VGLUT1 signal (see the representative

images below), suggesting that a neuronal overexpression approach is not suitable for addressing this question. We thus believe that it would be necessary to generate and characterize other KI mice possessing phosphomimetic variants of proteins that we isolated as hyperphosphorylated in the KI mice (e.g. dynamin-1, piccolo) or phosphodead variants of proteins isolated as hypophosphorylated (e.g. shank3). Such studies are beyond the focus of this manuscript, but, we have now noted this as a limitation of our study in the Discussion section (Page 18, line 603-607).

Fig. Neuronal overexpression of Dynamin1 WT or phosphomimetic constructs diminishes VGLUT1 puncta signals at axon-dendrite contact sites

Hippocampal cultured neurons were transfected with GFP or the indicated GFP-tagged Dynamin 1 construct at 7 days *in vitro* (DIV) and immunostained for VGLUT1, an excitatory presynaptic marker, and MAP, a dendrite marker, at 21 DIV. Arrowheads indicate contact sites between GFP-expressing axons and dendrites (MAP2-positive neurites). Scale bar: 5 μ m.

Minor

1. Fig. 1D: what does M stand for?

(Response) The highlighted “M”s indicate the detection of both adenine and cytosine bases at the same position within the same sample, which is expected for heterozygous samples. We have now clarified this in the figure legend. The double peaks at the mutation positions in the sequencing traces show that heterozygous mice have a mixture of the two residues.

2. Fig. 2A-C: is Vglut1 and PSD95 puncta number and overlap affected in TrkC KI mice?

(Response) The resolution of our confocal microscope images is insufficient for precise assessment of the puncta number and overlap of Vglut1 and PSD-95. Therefore, as an alternative precise assessment, we did EM analysis.

3. Fig. 2B: there appears to be a bimodal distribution in both the Vglut1 and PSD95 condition. Please explain.

(Response) We agree with the reviewer that there seemed to be a bimodal distribution. However, this was not due to immunofluorescent intensity differences between different experimental mice, but rather to variation between different slices of individual mice. We carried out additional

immunostaining and quantification to increase the sample number from n=3 to n=4, which shows a more representative distribution. In addition, as the reviewer suggested in the next comment, we performed nested analysis, and this shows that all TrkC KI mice show an increase in intensity (Appendix Fig. S3).

4. Fig. 2B-C: data points are ROIs; please perform a nested analysis so data points are mice. Similar for Fig. 3B.

(Response) We have now performed nested analysis for the experiments shown in Figures 2B,C and 3B and provide quantitative per mouse data in Appendix Fig. S3 and Fig. S5, respectively.

5. Fig. 2D: also show a representative image.

(Response) As the reviewer suggested, we have added a representative image in the new Figure 2D panel.

6. Fig. 2F: what is shown in graph, number of mice?

(Response) Each data point in Fig. 2F represents a separate hippocampal synaptosome sample, thus the total n=9 samples refer to 9 different synaptosome sample preparations. However, due to the low protein yield in a synaptosome preparation obtained from an individual mouse, we prepared each synaptosome sample from the pooled hippocampi of 6 mice of the same genotype. For better clarification, we combined Figures 2E and 2F into one panel and detailed this information in the figure legend.

7. Fig. 5C-D: how are the thresholds chosen?

(Response) The threshold based on $p < 0.05$ and $|FC| > 1.5$ is one of the standard ones used in many previous studies using volcano analysis. Further, as we described in the manuscript, the threshold allowed us to isolate altered phosphorylation sites by TrkC KI, ~ 98% of which also meet the confidence condition defined by false discovery rates of 0.1 in the hippocampus and 0.05 in the cortex, suggesting that the threshold would not include a significant number of false-positive data points.

8. Fig. 5C-D and 6A-B: what was rationale for including cortex instead of focusing on hippocampus as was done in the previous figures?

(Response) It is very challenging to validate hippocampal phosphoproteomic data in independent experiments because of the lack of appropriate antibodies. Therefore, as an alternative validation approach, we further performed cortical phosphoproteomics to see which phosphorylation changes detected in the hippocampus also occurred in the cortex. Then, we focused on

phosphorylation sites altered in both brain regions as confident data points for the subsequent bioinformatics analysis.

9. *The link between phosphoproteomics and behavior is not particularly strong. Can a statistical test be done for the SFARI associated proteins to determine whether these are over-represented in the dataset?*

(Response) It would be difficult to perform a statistical test. However, our phosphoproteomic data show that 15 (31.25%) of the 48 proteins that represent differential phosphorylation in both the hippocampus and the cortex are listed in the SFARI gene database. On the other hand, of all human genes (~30,000 genes), so far, a total of 1162 genes (3.87%) are listed in the SFARI gene database. Comparing between these percentages (31.25% vs 3.87%), it seems that our isolated molecules are likely to be over-represented in the dataset.

10. *As a suggestion, Figure 1 and 5 could be moved to supplemental figures; behavioral data could be condensed in one main and one supplemental figure.*

(Response) Considering this comment together with the reviewer 2 minor comment #1, we combined the previous Fig. 5AB (WB data for TrkC, PTP σ and NT-3) together with Figure 1 and kept this as a main figure (Fig. 1) because this mutant mouse line was newly generated in my lab. In addition, we integrated the previous Fig. 5C,D (hippocampal and cortical volcano plots in proteomic experiments) into the phosphoproteomic figures.

11. *The Discussion is well written and integrates the results in the existing literature. I missed discussion of previous studies assessing synaptic parameters in TrkC KO mice. Martínez et al., J Neurosci 1998 analyze synaptic density and presynaptic morphology in TrkC KO using EM; there may be additional studies. Similarities and differences with the TrkC KI will be interesting to discuss.*

(Response) We have added information about phenotypic differences between our TrkC KI mice and TrkC mutant mice lacking the tyrosine kinase domain or expressing non-catalytic TrkC isoforms (TrkC KO) to the Discussion section (page 15, line 485-488).

Referee#2:

In the present report, Khaled and colleagues generate and analyze a mouse knock-in model that disrupts the trans-synaptic interaction of TrkC with the post-synaptic receptor PTPsigma. In this elegant genetic model, the neurotrophic function of TrkC with the neurotrophin NT-3 is preserved, which allows to dissect the impact of the trans-synaptic adhesion on synapse organization, function and their behavioral consequences. With this model, the authors show that TrkC/PTPsigma interaction regulates the phosphoproteome of synapses, the excitatory presynaptic organization, the excitatory neurotransmission and alters social behavior. This is of major importance for several fields. The knock-in approach revealing multifunctional aspects of single proteins is of interest for most scientists interested in the structure/function

analysis of proteins. The analysis of TrkC functions is of broad interest for cell biologists. The analysis of the trans-synaptic adhesion system is of broad interest for molecular neurobiologists. The behavioral consequences of the mutation in relation to psychopathology is of broad interest for neuroscientists and clinicians.

However, several aspects of the work may request further experiments or major corrections prior to publication. The overall impression is that many topics have been investigated, but not enough to reach firm and reliable conclusions.

(Response) We are thankful for the reviewer's positive comments and many helpful suggestions to improve the manuscript. We hope that the additional data significantly improve the manuscript to strengthen the conclusions of this manuscript.

Major points:

1/ VGLUT1 expression / distribution

The authors provide immunofluorescence data that strongly suggest a global increase of expression of VGLUT1 both in-vivo and in-vitro in the TrkC-KI. Based on an immunoblot on synaptosomes that shows no difference of expression of VGLUT1, the authors conclude that TrkC-KI leads to a relocation of VGLUT1 but not to a change of expression level. There is a need for additional experiments on this point to reach a conclusion that is fully supported by the data. Indeed, the authors immunostainings reveal an increase of signal that is difficult to reconcile with a relocation of the proteins. Furthermore, the images from cultured neurons do not suggest a change of distribution. Therefore, either the authors provide additional data showing direct evidence for a change of trafficking of VGLUT1 or they have to show a consistent set of data showing the impact of TrkC-KI on the expression level of VGLUT1. In general, a set of immunoblots of an array of synaptic proteins on homogenates of hippocampus and/or cortex may also be helpful. For PSD-95 the dataset is more coherent because the authors show a more local change of expression in the stratum radiatum layer of the hippocampus that can be explained by the increased PSD size measured in electron microscopy only in that layer. This increase may escape a global immunoblot analysis for the whole hippocampus.

(Response) Indeed, as the reviewer mentioned, we were also initially puzzled with the VGLUT1 immunofluorescent signal increase in the absence of a corresponding change in the synaptosome expression level. Thus, we carried out further proteomic analysis (Fig. 5A,B in the revised manuscript) showing no increase in VGLUT1 protein level and electron microscopy analysis (Fig. 3) showing similar numbers of synapses as well as synaptic vesicles in WT and KI samples, which together support our immunoblot data showing no difference in the expression level of VGLUT1. In addition, our electron microscopy analysis showed that, while synapse vesicle number is similar, they are more clustered in TrkC KI mice, as measured in the synaptic vesicle distance analysis, providing an explanation for the signal increase observed in immunofluorescence data.

To strengthen this argument, and in accordance with the reviewer's suggestion, we have added new immunoblot data for VGLUT1 and PSD-95 total protein levels in hippocampal total homogenates (new panel Fig. 2F), and these data show no significant difference in the total expression of these proteins between WT and TrkC KI mice. Furthermore, we carried out immunoblot analysis of the expression VGLUT1 and PSD-95 using cortical synaptosome samples, and these data also showed no significant difference in expression in WT and TrkC KI samples (data not shown in the revised manuscript).

Of note, we would also like to clarify that we do not suggest a change in relocation, distribution or trafficking of VGLUT1, especially not in the context of extra-synaptic VGLUT1 trafficking to the synapse. Rather, based on our electron microscopy analysis, we are only suggesting a tighter clustering of the already present synaptic vesicles. This is also in accordance with TrkC KI cultured neurons showing no change in distribution of VGLUT1, yet still an increase in signal intensity.

2/ Ultrastructural changes at excitatory hippocampal synapses of TrkC KI

The authors provide a thorough analysis of the TrkC KI compared to WT using transmission electron microscopy.

- The measurement of synapse density is correct in my opinion and comforts the idea that synapse density is unchanged.

- The measurement of PSD size seems appropriate and displays a local increase in stratum radiatum. A further analysis of this phenotype may be of interest. Are other post-synaptic proteins and receptors regulated in this region? What is the functional impact?

(Response) To address how glutamate receptors are regulated at the hippocampal CA1 st. rad. in TrkC KI mice, we performed two additional electrophysiology experiments using whole cell patch-clamp recording from hippocampal CA1 pyramidal cells: (1) we assessed AMPAR- and NMDAR-mediated EPSCs evoked by Schaffer collateral (SC) stimulation, and (2) we assessed AMPAR-mediated miniature EPSCs (mEPSCs).

In the evoked EPSC experiments (Fig. 4A-C), we first quantified the AMPAR/NMDAR ratio and found that this is lower in TrkC KI synapses than in WT synapses, suggesting that TrkC KI synapses may show impairment of synapse maturation. We further quantified the CV-NMDAR/CV-AMPA ratio, which allows us to measure silent synapses (NMDAR-positive but AMPAR-negative synapses) and found that this is also lower in TrkC KI synapses, suggesting that there is a larger number of silent synapses in TrkC KI mice than in WT mice. Given our EM data showing no difference in total excitatory synapse number in the st. rad of TrkC KI and WT mice, the evoked EPSC data also suggest that TrkC KI mice have a smaller number of active synapses (NMDAR- and AMPAR-positive synapses) than WT mice.

We next performed AMPAR-mediated mEPSC experiments to assess the function of active synapses by quantifying mEPSC frequency and amplitude (Fig. 4D-F). Very interestingly, we observed a significant increase in mEPSC frequency in TrkC KI synapses. On the other hand, mEPSC amplitude was comparable between TrkC KI and WT synapses. These data suggest that active synapses in TrkC KI mice may show increased glutamate releasing sites and/or increased release probability without changing postsynaptic AMPARs.

Our data about AMPAR-mediated fEPSPs already revealed that TrkC KI SC-CA1 synapses show a significant increase in fEPSP slopes in I/O curves and significant enhanced paired-pulse facilitation, which suggests impaired release probability (Fig. 4G-J). Integrating this with our new mEPSC results suggests that active synapses in TrkC KI mice may have more releasing sites, which is consistent with increased fEPSP slopes as well as increased mEPSC frequency, but impaired probability of glutamate release from the releasing sites, especially under action potential-dependent release conditions.

Given these new results, we revised the Results section describing the electrophysiology experiments to conclude that TrkC KI substitution increases silent synapses (Page 7, line 227-230) and causes aberrant active synapses with abnormally enhanced basal synaptic transmission (presumably due to increased glutamate releasing sites) but lower release probability than WT active synapses (Page 8, line 251-255). We further added a discussion that such aberrance of active synapses in TrkC KI mice might be compensatory to delayed synapse maturation and increased silent synapses (Page 15-16, line 519-538).

- The measurement of synaptic vesicle distribution and density is more problematic in my opinion, because it is now broadly accepted that chemical fixation methods alter SV distribution in the terminals (see for instance Siksou, Jneurosci, 2007; Siksou, EJN, 2013; Korogod, elife, 2015). Alternatively, high pressure freezing provide a much better tissue preservation and quality for this type of analysis. Data such as the number of docked/primed SVs is accessible for instance. Hence, the present data suggests that the SV cluster is more compact, but it is likely a joint consequence of relevant cellular alterations that change the

dynamics of SV mobility induced by the KI together with irrelevant crosslinking induced by the chemical fixation. The authors must take this into account to revise their conclusion or perform high pressure freezing experiments to produce a complementary data set helping to draw firm conclusions.

(Response) We agree that high-pressure freezing (HPF) experiments are superior for precise assessment of SV density and distribution, especially near the presynaptic membrane, as Korogod, Elife, 2015 has shown. Unfortunately, our institute does not have the equipment for HPF, therefore we had to use a traditional method based on chemical fixation. Indeed, in some synapses (mostly TrkC KI synapses), the assessment of SV density and distribution near presynaptic membranes was difficult because these synapses showed a dark cloudy structure, which was likely to be tightly clustered SVs near the presynaptic membrane, suggesting that we may have underestimated the number of SVs. Considering the reviewer's valuable comment and the qualitative observation of a dark cloudy structure in some synapses in our chemically-fixed EM samples, we added a discussion of this limitation of our EM study and the importance of HPF experiments in the Discussion section (Page 18, line 593-601).

- In figure 3E, SV density measurements are displayed by number of SV/bouton, this might induce a bias as the measurements were done on single thin sections and not by serial sections or 3D tomography. Instead, the area of boutons where SVs are counted should be used for normalization. The results will be displayed as SV number per μm^2 of bouton area.

Altogether, the EM analysis is of high potential interest with respect to the observation of VGLUT1 and PSD95 expression and phosphor-proteomic alterations, but the methodology should be revised to reach reliable conclusions.

(Response) We thank the reviewer for pointing this out. We have re-analysed the images and changed the graphs from "SVs / bouton" to represent "SVs / μm^2 of bouton area".

3/ Proteomics analysis

The authors provide 2 proteomics datasets comparing TrkC KI and WT. A global quantification of proteins of the hippocampal synaptosomes and a phosphor-proteomic analysis of the same sample using labeling methods.

- The global assay shows no significant difference between genotypes. This is most likely due to the fact that synaptosome preparations are quite bulk compared to the changes that occur only at some excitatory synapses in this KI model. I'm not sure that it is wise to conclude that no changes of protein amounts occur in this mouse model from this experiment on bulk synaptosomes.

(Response) We agree with the reviewer's comment because bulk synaptosomes contain the components of not only CA1 synapses but also those of the other hippocampal synapses such as mossy fiber-CA3 synapses, which we have not addressed in this study. Further, for proteomic

experiments, we used crude, but not pure, synaptosomes, which may only allow rough assessment of synaptic protein abundance, as the reviewer suggested. Therefore, we removed the conclusion that no change in protein amounts occurs in TrkC KI synapses, and rather conclude that the synaptic phenotypes in TrkC KI mice are unlikely to be due to altered synaptic expression of TrkC and PTP σ or any other known molecules involved in synaptic organization and function (page 8, line 263-267).

- The phosphor-proteome dataset shows some potentially interesting differences in the phosphorylation of proteins involved in synapse biology and may have a link with psychiatric behavioral dimensions. Yet, a validation of some of these potentially interesting targets with independent methods is needed to strengthen the finding. A connection between altered phosphorylation of some proteins and the phenotypes of SV dynamics could be investigated to clarify some opened questions of the current study.

(Response) It is challenging to validate phosphoproteomics datasets by standard western blot (WB) experiments because of the lack of antibodies that detect phosphorylation of the identified sites. We used an alternative approach based on Phos-tag WB (PMID: 26676139). Phos-tag traps phosphorylated proteins and allows us to distinguish between the phosphorylated and non-phosphorylated forms of the protein of interest. Of the proteins isolated in our phosphoproteomics experiment, we selected ITPKA because it had one of the highest fold changes in hippocampal samples and all phosphopeptides of ITPKA in TrkC KI mice were hyperphosphorylated compared to WT. Phos-tag WB using an anti-ITPKA antibody confirmed the hyperphosphorylation of ITPKA in TrkC KI mice (Appendix Fig. S6).

In addition, our WB experiments show no change in the phosphorylation of NT-3 signaling molecules, consistent with our phosphoproteomic results (Appendix Fig.S8).

To demonstrate how altered phosphorylation levels of the synaptic proteins identified here alters pre- or postsynaptic properties, we attempted a cell-biological experiment to mimic the phosphorylation changes in dynamin-1 that we observed in our phosphoproteomic experiments (Fig. 5: S853;S857 hyperphosphorylation in both hippocampal and cortical samples as well as S774D;T776D hyperphosphorylation in the hippocampal samples). We therefore induced neuronal overexpression expression of WT or phosphomimetic dynamin-1 (S853D;S857D, S774D;T776D and S774D;T776D;S853D;S857D) and then assessed VGLUT1 puncta because TrkC KI cultured neurons show an increase in VGLUT1 signals. However, we found that overexpression of any dynamin1 construct (including WT) resulted in significant reduction of VGLUT1 signal (see the representative images below), suggesting that a neuronal overexpression approach is not suitable for addressing this question. We thus believe that it would be necessary to generate and characterize other KI mice possessing phosphomimetic variants of proteins that we isolated as hyperphosphorylated in the KI mice (e.g. dynamin-1, piccolo) or phosphodead variants of proteins isolated as

hypophosphorylated (e.g. shank3). Such studies are beyond the focus of this manuscript, but, we have now noted this as a limitation of our study in the Discussion section (Page 18, line 603-607).

Fig. Neuronal overexpression of Dynamin1 WT or phosphomimetic constructs diminishes VGLUT1 puncta signals at axon-dendrite contact sites

Hippocampal cultured neurons were transfected with GFP or the indicated GFP-tagged Dynamin 1 construct at 7 days *in vitro* (DIV) and immunostained for VGLUT1, an excitatory presynaptic marker, and MAP, a dendrite marker, at 21 DIV. Arrowheads indicate contact sites between GFP-expressing axons and dendrites (MAP2-positive neurites). Scale bar: 5 μ m.

- Some important information is missing to allow for a proper evaluation of both datasets. Like for instance how many proteins were identified, Criteria for inclusion in the differential quantification, coverage of the syngo database...

(Response) We added the number of proteins identified by each proteomic or phosphoproteomic experiment into each Volcano plot figure. We also added the criteria for inclusion in the differential quantification ($p < 0.05$ and $|FC| > 1.5$ in volcano plot) in the legend for Figure 5C,D. Regarding the coverage of the SynGo database, we added the following description into the legend for Figure 5H.

Page 48, line 1387-1389: “18 of the altered 48 genes have Biological Processes annotation in the SynGO analysis. 10 of 208 Biological Processes terms were significantly enriched at 1% FDR (testing terms with at least three matching input genes).”

4/ Behavioral assessment

The authors provide very interesting phenotypes from the assays performed. Yet a thorough SHIRPA set of tests would be nice to make sure that no major phenotype escaped the attention of the investigators.

(Response) Due to time constraints and a lack of equipment and resources, we are unable to carry out a full SHIRPA screen. However, we have carried out analysis for some of the general mouse observations and transfer behaviors (Appendix Fig.S10), which represent a portion of the tests included in a SHIRPA screen. In the analysis, TrkC KI mice did not show a significant difference in any analyzed parameters, except for the increased freezing and jumping behaviours already reported in the main manuscript and Figs. 6 and 7.

5/ Altogether, it is unclear at the end of the reading how the authors link the mutation they induced to the alteration of VGLUT1, PSD95, synapse ultrastructure, synapse phosphor-proteome and mouse behavior. While some interesting but incomplete evidence are provided, causal links are often missing.

(Response) By addressing the reviewer 2's major comments #1-#4, we think that in the revised manuscript, it became clearer how the TrkC KI mutation is linked to synaptic and behavioral phenotypes. On the other hand, the investigation of causal links remains challenging, especially to demonstrate which phosphorylation change causes synaptic and/or behavioral phenotypes. As mentioned above, we performed neuronal overexpression experiments to investigate whether and how dynamin1 hyperphosphorylation affects VGLUT1 signal. Through this experiment, we found that to address causal links, it would be necessary to generate and characterize additional KI mutant mice expressing phosphomimetic or phosphodead variants of proteins of interest. Therefore, we added a discussion about the limitation of our current study and what future studies would be necessary to address causal links (Page 18, line 603-607).

Minor points:

1/ Grouping the KI validations in figure 1 may be better that having some aspects in other figures like Fig 5AB or Fig S4

(Response) We agree with the reviewer's suggestion. We thus merged the previous Fig. 5AB (WB data for TrkC, PTP σ and NT-3) into Figure 1.

2/ Positive control for NT3 signaling

While the KI mice seem to be unaffected for NT3 signal transduction, it seems that the article lacks an assay that measures this in a quantitative and dynamic way.

(Response) For quantitative assessment of NT-3 signaling activity, we performed new WB experiments for phospho-MAPK1/3, phospho-AKT and phospho-PLC γ (three signaling molecules downstream of NT-3) in hippocampal homogenate samples and found no significant differences between TrkC KI and WT samples (Appendix Fig. S8C).

To check the dynamics of NT-3 signal transduction, it would be necessary to perform western blotting for NT-3 downstream signals after exogenous NT-3 treatment. Although such experiments are interesting, the current manuscript focuses on endogenous NT-3 signal transduction in TrkC KI mice.

Dear Dr. Takahashi,

Congratulations on a great revision! Overall, the referees have been positive. However, referee 2 has a remaining concern that we ask you to (non-experimentally) address in a new revision. When you submit your revised version, please also take care of the following editorial items and add this also to your point-by-point response:

1. Please remove the figures from the main manuscript.
2. Please move the Data Availability section to the end of the Methods section.
3. Please remove the author contribution section from the main manuscript.
4. Please merge the supplemental methods with methods in main manuscript text; nomenclature in legends should be corrected to "Appendix Figure S1" etc. and "Appendix Table S1" etc.
5. Thank you for providing source data. The data for figure 5 should be deposited externally and mentioned in the data availability section.
6. Please provide a reagent table.
7. Please resize the synopsis image to 550 pixels wide by 200-440 pixels high. When you adjust the size, please ensure the text is still legible and make bigger if necessary.
8. All figures should be mentioned in the main manuscript. Please include a call out to figure 4F in the main manuscript.
9. Please note that the exact p values are not provided in the legends of figures 2b-c; 3c, f; 4g; 6a, c; 7b, d.
10. Please indicate the statistical test used for data analysis in the legends of figures 5a-d, g.
11. Please note that the error bars are not defined in the legends of figures 4b-c, e-j.
12. Please note that the asterisk/ hash are not defined in the legend of figure 1b. This needs to be rectified.

Thank you for the opportunity to consider your work for publication. I look forward to your revision.

Warm wishes,
Kelly

Kelly M Anderson, PhD
Editor, The EMBO Journal
k.anderson@embojournal.org

Referee #1:

I appreciate the strong effort of the authors to add experimental data to their manuscript. These additions, together with the clarifications added in the point-by-point reply and text, address all my concerns and have further improved their interesting study. The paper is ready for publication in my view.

Referee #2:

In the present revised report, Khaled and colleagues corrected their study on a mouse knock-in model that disrupts the trans-

synaptic interaction of TrkC with the post-synaptic receptor PTPsigma according to referees' comments. By lack of time, I only focused my present review on the most salient aspects. Additional careful reading for mistakes and typos should be done independently. In my opinion, the authors have taken seriously the comments of all referees to improve the report and align the conclusions to the observations taking into account potential limitations of the methods. However, one aspect of the work may request further experiments or major corrections prior to publication.

Major point :

1/ VGLUT1 expression / distribution

At this stage, the authors convincingly show that while bulk measures of VGLUT1 expression show no difference, local measures using immunofluorescence do. This may suggest that the stoichiometry of VGLUT1 molecules per synaptic vesicle (SV) is changed without any major change in VGLUT1 expression level. This may have consequences on SV super-pool size, SV clustering and release probability as VGLUT1 c-terminal proline rich domain was shown to interact with endophilinA1 with consequences on these phenomena (see PMID: 15118123; PMID: 21435559; PMID: 23581566; PMID: 31663854). It is therefore important to show whether there is a changed ratio to some SV proteins (synapsin1/2; Synaptophysin; SV2; Synaptotagmin1) that may explain some of the other phenotypes observed in electron microscopy and electrophysiology.

Authors' response to the referees' comments and editorial items for the manuscript **EMBOJ-2024-116855R1** of Khaled et al., **"The TrkC-PTP σ complex governs synapse maturation and angiogenic avoidance via synaptic phosphorylation"**

Referee #1:

I appreciate the strong effort of the authors to add experimental data to their manuscript. These additions, together with the clarifications added in the point-by-point reply and text, address all my concerns and have further improved their interesting study. The paper is ready for publication in my view.

Response: Thank you kindly for the positive evaluation of our revised manuscript.

Referee #2:

In the present revised report, Khaled and colleagues corrected their study on a mouse knock-in model that disrupts the trans-synaptic interaction of TrkC with the post-synaptic receptor PTP σ according to referees' comments. By lack of time, I only focused my present review on the most salient aspects. Additional careful reading for mistakes and typos should be done independently. In my opinion, the authors have taken seriously the comments of all referees to improve the report and align the conclusions to the observations taking into account potential limitations of the methods. However, one aspect of the work may request further experiments or major corrections prior to publication.

Major point:

1/ VGLUT1 expression / distribution

At this stage, the authors convincingly show that while bulk measures of VGLUT1 expression show no difference, local measures using immunofluorescence do. This may suggest that the stoichiometry of VGLUT1 molecules per synaptic vesicle (SV) is changed without any major change in VGLUT1 expression level. This may have consequences on SV super-pool size, SV clustering and release probability as VGLUT1 c-terminal proline rich domain was shown to interact with endophilinA1 with consequences on these phenomena (see PMID: 15118123; PMID: 21435559; PMID: 23581566; PMID: 31663854). It is therefore important to show whether there is a changed ratio to some SV proteins (synapsin1/2; Synaptophysin; SV2; Synaptotagmin1) that may explain some of the other phenotypes observed in electron microscopy and electrophysiology.

Response: We would like to thank the referee for the overall positive evaluation of our manuscript and providing critical comments. Considering the above major point, we revised the Discussion section to highlight the importance of a future study that will address VGLUT1

stoichiometry per SV, as shown below.

Page 18, line 598 - 603 in the PDF file of the manuscript:

“First, although our EM data suggest that the increased signal of VGLUT1 in the IHC is likely to result from the enhanced clustering of SVs, it remains possible that this might also result from the altered stoichiometry of VGLUT1 per SV. To address this possibility, it is important to conduct a biochemical analysis of highly-purified SVs (Takamori et al, 2006) and/or a super-resolution microscopy imaging of isolated, single SVs (Upmanyu et al, 2022).”.

Accordingly, the following references were added into the Reference list.

Takamori S, Holt M, Stenius K, Lemke EA, Grønborg M, Riedel D, Urlaub H, Schenck S, Brügger B, Ringler P *et al* (2006) Molecular anatomy of a trafficking organelle. *Cell* 127: 831-846

Upmanyu N, Jin J, Emde HV, Ganzella M, Bosche L, Malviya VN, Zhuleku E, Politi AZ, Ninov M, Silbern I *et al* (2022) Colocalization of different neurotransmitter transporters on synaptic vesicles is sparse except for VGLUT1 and ZnT3. *Neuron* 110: 1483-1497 e1487

In addition, considering the reviewer’s comment about endophilin A1, we revised the Discussion section as shown in red below by adding the very recently published paper (Imoto *et al*, 2024, *EMBO J.*) because this paper shows that dynamin-1 binds to endophilin A1, depending on dynamin C-terminal phosphorylation, in order to regulate SV endocytosis, consistent with our phosphoproteomic experiment that detected the altered phosphorylation levels of the dynamin-1 C-terminal region (Dnm1_S853;S857: Fig. 5C, 5D, 5F). Accordingly, we further added the references that describe the phosphorylation of other synaptic proteins, as shown in red below.

Page 15, line 505 - 513 in the PDF file of the manuscript:

“Many different types of presynaptic molecules **and their phosphorylation** are known to underlie the regulation of SV clustering and SV exo-/endocytotic recycling for neurotransmitter release, such as synapsin I, an SV-associated protein (**Bonanomi et al, 2005**; Shupliakov *et al*, 2011), bassoon and piccolo, active zone proteins (Garner *et al.*, 2000; Gundelfinger *et al.*, 2015), the SNARE protein complex, intracellular membrane fusion machinery (Sudhof & Rothman, 2009; **Turner et al, 1999**), Caps-1

(Cadps), a synaptic vesicle priming protein (Jockusch et al., 2007; Nojiri et al, 2009), and dynamin-1, a membrane severing protein involved in clathrin-dependent endocytosis (Ferguson & De Camilli, 2012; Imoto et al, 2024; Tomizawa et al, 2003).”

Accordingly, the following references were added into the Reference list.:

Bonanomi D, Menegon A, Miccio A, Ferrari G, Corradi A, Kao HT, Benfenati F, Valtorta F (2005) Phosphorylation of synapsin I by cAMP-dependent protein kinase controls synaptic vesicle dynamics in developing neurons. *J Neurosci* 25: 7299-7308

Imoto Y, Xue J, Luo L, Raychaudhuri S, Itoh K, Ma Y, Craft GE, Kwan AH, Ogunmowo TH, Ho A et al (2024) Dynamin 1xA interacts with Endophilin A1 via its spliced long C-terminus for ultrafast endocytosis. *EMBO J* 43: 3327-3357

Nojiri M, Loyet KM, Klenchin VA, Kabachinski G, Martin TF (2009) CAPS activity in priming vesicle exocytosis requires CK2 phosphorylation. *J Biol Chem* 284: 18707-18714

Tomizawa K, Sunada S, Lu YF, Oda Y, Kinuta M, Ohshima T, Saito T, Wei FY, Matsushita M, Li ST et al (2003) Cophosphorylation of amphiphysin I and dynamin I by Cdk5 regulates clathrin-mediated endocytosis of synaptic vesicles. *J Cell Biol* 163: 813-824

Turner KM, Burgoyne RD, Morgan A (1999) Protein phosphorylation and the regulation of synaptic membrane traffic. *Trends Neurosci* 22: 459-464

Dear Dr. Takahashi,

Congratulations on an excellent manuscript. I am pleased to inform you that your manuscript has been accepted for publication in The EMBO Journal. Thank you for your comprehensive response to the referee concerns and for providing detailed source data. It has been a pleasure to work with you to get this to the acceptance stage.

I will begin the final checks on your manuscript before submitting to the publisher next week. Once at the publisher, it will be about 3 weeks for your manuscript to be published online. As a reminder, the entire review process including referee concerns and your point-by-point response will be available to readers.

I will be in touch through the final editorial process until publication. In the meantime, I hope you find time to celebrate!

Warm wishes,
Kelly

Kelly M Anderson, PhD
Editor, The EMBO Journal
k.anderson@embojournal.org
